**Subject Category:**

Mathematics

evolution/mathematical modelling/theory of computing

evolutionary dynamics, sexual reproduction, population genetics, potential games

**Author for correspondence:**

Omer Edhan
e-mail: omeredhan.idan@manchester.ac.uk

# Making the most of potential: potential games and genotypic convergence

Omer Edhan[1], Ziv Hellman[2] and Ilan Nehama[2]

[1]Department of Economics, University of Manchester, Manchester, UK
[2]Department of Economics, Bar-Ilan University, Ramat Gan, Israel

  OE, 0000-0002-4441-3304; ZH, 0000-0002-2624-0577;
IN, 0000-0002-4152-4113

We consider genotypic convergence of populations and show that under fixed fitness asexual and haploid sexual populations attain monomorphic convergence (even under genetic linkage between loci) to basins of attraction with locally exponential convergence rates; the same convergence obtains in single locus diploid sexual reproduction but to polymorphic populations. Furthermore, we show that there is a unified theory underlying these convergences: all of them can be interpreted as instantiations of players in a potential game implementing a multiplicative weights updating algorithm to converge to equilibrium, making use of the Baum–Eagon Theorem. To analyse varying environments, we introduce the concept of 'virtual convergence', under which, even if fixation is not attained, the population nevertheless achieves the fitness growth rate it would have had under convergence to an optimal genotype. Virtual convergence is attained by asexual, haploid sexual and multi-locus diploid reproducing populations, even if environments vary arbitrarily. We also study conditions for true monomorphic convergence in asexually reproducing populations in varying environments.

## 1. Introduction

One of the central questions of evolutionary theory has long been identifying conditions for asymptotic convergence to fixation on a monomorphic population. The classical example of such a result is the simplest case of asexual reproduction without mutation (e.g. bacteria reproducing in a petri dish) in which a version of the fundamental theorem of natural selection obtains: the mean fitness of the population, which follows the dynamic of the replicator equation, increases monotonically, leading to asymptotic fixation to a monomorphic population consisting of an optimal genotype with respect to the fitness environment.

Even this strong result, however, fails to hold once one considers arbitrarily varying fitness environments over time, even

in asexually reproducing populations; in sexually reproducing populations the matter is more complicated still. In this paper, we consider the general question of genotypic convergence of populations implementing various reproductive strategies under conditions of both fixed and varying environments. To this end, we also introduce a concept that we term 'virtual convergence', applying ideas originally developed for the study of algorithms.

In greater detail, we consider here three discrete time population reproductive strategies: asexual, haploid sexual and diploid sexual. The relevant state spaces for all of these is a polytope $\Theta$. In the asexual case $\Theta = \Delta(\Gamma)$, the space of probability distributions over the set $\Gamma$ of possible genotypes, what is of interest is tracing over time the relative frequency of the genotypes. In the sexual cases, the focus instead is on the relative frequency in the population of alleles at each locus; if there are $m$ loci with $k + 1$ alleles per locus, the polytope of interest is $\Theta = \Delta_1^k \times \cdots \times \Delta_m^k$.

The dynamics considered will in general be describable by a transformation $T : \Theta \to \Theta$. That is, if the population is at state $\theta \in \Theta$ at time $t$, under the model it will be in state $T(\theta)$ at time $t + 1$. The main matter studied is then the asymptotics of the trajectory defined by $T^n(\theta)$, starting from any $\theta$, as $n$ increases. If for each initial point $x \in \Theta$ there is a point $y \in \Theta$ such that $\lim_{n \to \infty} T^n(x) = y$ then the dynamic converges polymorphically; if $y$ is a point distribution in $\Delta_i^k$ for each $1 \leq i \leq m$ then the convergence is monomorphic.

## 1.1. Fixed environments and convergence in potential games

The first question we consider asks which of these dynamics is guaranteed to converge, either monmorphically or polymorphically, when environments are fixed and unchanging over time, and we show that the asexual replicator dynamic, the sexual haploid dynamic—even under genetic inheritance linkage between loci—and the single-locus diploid dynamic all converge.

Furthermore, we provide a unified explanation for the convergence of all of these dynamics in the discrete time setting: all of them may be considered to be manifestations of potential games in which the players monotonically increase the potential payoff.

Potential games (introduced in [1]) satisfy the property that the incentives of the players to change strategies are all captured in one global potential function—the name is inspired by the concept of a potential in physics—that is common to all the players. Significantly, this introduces the possibility that the players can together 'climb' the potential to attain (at least locally) optimal payoffs.

The key to several of our convergence results is due to theorem 3.1, in which we show that, whenever a set of players playing a potential game each implement the multiplicative weights updating algorithm in sequentially choosing their actions, the play will always converge to a fixed point that is a Nash equilibrium. Furthermore, crucially the dynamic always implements a monotonic climb of the potential payoff, even though there is no coordinating element at all to the updating of the players, each of whom updates based on the private information of the stage payoff received without explicitly taking into account the payoffs and updated distributions of the other players.

This result ultimately depends on an application of the Baum–Eagon inequality (see appendix A), which was originally intended for application to the study of hidden Markov models, but has proved to be valuable for the study of discrete time dynamics, where the standard tools of continuous time gradient climbing, which depend on partial derivatives, are not available.

This is especially pertinent to our study of convergence in the sexual haploid model, where the dynamic can be described as an identical interests game being played by the loci, with the objective being identifying an optimal genotype; the theorem shows that the replicator dynamic conducted independently among the alleles at each locus, which is the essence of the sexual reproduction model, is guaranteed to converge. The exception to all this is the multi-locus diploid model under linkage disequilibrium, where the disequilibrium term prevents application of the Baum–Eagon inequality, and in fact it has long been known that convergence under that model is not guaranteed.

A further advantage of undergirding the fixed environment theorems by appeal to the Baum–Eagon Theorem is that it enables us to make use of theorems from [2] to obtain finer resolution insights into the dynamic paths followed by populations along the way towards convergence. This includes the fact that surrounding each pure Nash equilibrium there exists a basin of attraction, and even more strongly a basin of attraction that is exponentially stable. This implies that an observer following a path through the state space (including that of any potential game in which the players are implementing the polynomial multiplicative weights updating algorithm) will for a long time register relatively small increase in mean payoff until the path enters the exponential basin of attraction, at which point an acceleration will be noted with exponentially fast convergence to a fixed point of local maximal mean payoff.

## 1.2. Varying environments and virtual convergence

When environments vary, sufficiently wildly varying environments from one time period to the next can make it impossible for the dynamic to converge to any single population state in $\Theta$. To contend with this we introduce here a new concept of 'virtual convergence'. This is defined using tools borrowed from computer science and introduced to the population genetics literature in the past decade, which involve regret minimization algorithms. The metaphor often used to describe this approach is that of selecting an action with respect to varying payoff functions in subsequent time periods based on advice offered by a collection of experts. The objective is attaining asymptotically the payoff that would have been achieved had one followed from the start the advice of the best expert in hindsight in every time period; a no regret algorithm achieves this objective.

In the evolutionary setting, the analogues of the experts of the previous paragraph are genotypes and the payoffs are fitness values. The question then becomes: is it the case that, no matter what sequence of environments and hence fitness values is realized, the reproducing population asymptotically attains the mean fitness that is the growth rate that it would have achieved had it been comprised from the start monomorphically by the optimal-in-hindsight genotype? If yes, then we say that virtual convergence is attained.

With these definitions, we study here modes of convergence for the asexual, haploid sexual and diploid sexual reproduction, variously under independence of inheritance between loci as well as genetic linkage, fixed fitness and varying fitness conditions.[1] A summary of some of the results appears in table 1.

As can be seen in the summary, all of the reproduction models studied here attain virtual convergence, no matter how wildly environments vary. They attain this by exploiting the regret minimization aspect of the multplicative weights updating algorithm.

In a sense, it can be said that the reproductive processes studied here are nearly as opportunistic as they can be. When the environment is fixed, they will converge to local optima as represented by Nash equilibria. In some cases, when the environment is sufficiently regular (i.i.d. or stationary ergodic), the information inherent in the process of the changing environment can be extracted to yield optimal results (see proposition 4.1).

Virtual convergence captures the capacity of populations to minimize regret in hindsight, thus ensuring that the asymptotic growth per time period equals the rate that could have been achieved by selecting ahead of time the genotype of highest growth rate. Significantly, this shows that even in the most extremely arbitrarily varying environments the reproductive processes manage to do the best they can under the circumstances. Conceptually this underscores how impressively efficient the reproductive processes that have evolutionarily emerged are. These results may also enable better predictions of the outcomes of evolutionary processes in future research efforts.

## 1.3. Non-arbitrarily varying environments

The gap between fixed environments and entirely arbitrarily varying environments is large. The subject of convergence when environments vary in a structural way is explored here only with respect to the asexual replicator model, where we show that convergence to a monomorphic population is guaranteed under ergodically varying environments and under a broader property we introduce that we call one-step-ahead superiority.

## 1.4. Literature review

Our theorem on the convergence of potential games to pure Nash equilibria when all players implement the MWU algorithm (theorem 3.1) is equivalent to a theorem in [3] on convergence to pure Nash equilibria in congestion games. Our result provides an independent proof for potential games, in which form it readily applies to the evolutionary contexts that are the focus of this paper.

The results on asexual reproduction in fixed fitness environments mentioned in §4 are well-known and standard in the literature. Our most significant new contribution here is theorem 4.4, in which we introduce the concept of asymptotically one-step-ahead superior on average genotypes, showing that under the asexual replicator dynamic if the population contains such a genotype then it will converge to that genotype under arbitrarily varying environments.

---

[1]In all models in this paper, generations are discrete and non-overlapping, populations are infinite, and no mutation, migration or genetic drift is included in the models.

**Table 1.** Summary of modes of convergence. The code for the simulation generating figure 1 can be found in the electronic supplementary material.

| reproduction | fixed fitness convergence | varying fitness convergence | section |
|---|---|---|---|
| asexual | monomorphic | virtual | section 4 |
| haploid | monomorphic | virtual | section 5 |
| single locus diploid | polymorphic | virtual | section 6.1 |
| multi-locus diploid | virtual | virtual | section 6.2 |

That the Baum–Eagon inequality sheds light on the convergence properties of haploid reproduction, as finds expression here in lemma 5.1, has long been noted by many researchers, stretching all the way back to [4] (see also, among others, [5–7]). However, many of those sources fail to relate the application of the Baum–Eagon inequality to the underlying common interests game and most significantly to the Nash equilibria of that game, which is accomplished here in theorem 5.2. We also make use of results from [2] (the follow-up paper to [4]) to study details regarding basins of convergence to the Nash equilibria in the haploid dynamic (theorem 5.4).

Further contributions in this paper in the haploid reproduction setting are the monomorphic convergence results under genetic linkage (theorem 5.8) and the virtual convergence under varying environments (theorem 5.9).

Convergence to mixed strategy Nash equilibria in the single-locus diploid setting under fixed fitness, as noted in §6.1, is well-established (e.g. [8]); as in the haploid case, we contribute here insights into the structure of the basins of convergence (theorem 6.1), and aspects of virutal convergence in varying environments (theorem 6.4).

Two papers with some overlap with the topics in this paper are [9,10]. The first explores natural selection in stochastically varying environments. In such an environment it is often assumed that organisms with state dependent strategies have an evolutionary advantage that increases with capacity to detect environmental changes. In [9], it is shown, however, that counterintuitively there are models in which lower accuracy in detecting changes actually leads to higher growth.

In [10], Cheong *et al.* study cooperation between populations in periodically varying environments, especially prisoners' dilemma situations with periodically varying payoffs. The periodic payoff component is added in such a way that the behaviour of a given population depends both on its own payoffs and the payoffs of its opponent. This creates an opportunity for cooperation under the replicator dynamics.

The results of those papers are complementary to our results on evolutionary reproductive algorithms under varying environments. We leave it to future research to study how virtual convergence may shed light on situations of state dependent reproductive strategies and the evolution of cooperation.

# 2. Basic models and notation

## 2.1. Simplices

For an integer $m$, $\Delta^m$ denotes the standard finite dimensional simplex over $m + 1$ points. For a finite set $\Gamma$, $\Delta(\Gamma)$ denotes the collection of probability mass functions over the elements of $\Gamma$. We will denote the subset of $\Delta(\Gamma)$ consisting of distributions with support on one element of $\Gamma$ alone by $\Delta_1(\Gamma)$, and the element of $\Delta_1(\Gamma)$ placing all support on $g \in \Gamma$ will be denoted by $1_g$.

## 2.2. Potential games

Let $I$ be a finite set of $m$ players. Associate with each player $i$ a finite set of actions $A_i$. Denote $A = A_1 \times \cdots \times A_m$, and the cross product of all action sets except from $i$ by $A_{-i}$. A game is defined by a payoff function $u : A \to \mathbb{R}^m$. The projection of the payoff function to the payoff of player $i$ is denoted $u_i(a_i, a_{-i})$. Payoff functions extend in the obvious multi-linear manner to payoff functions of mixed strategies.

An identical interests game is a game satisfying the property that $u_i(a) = u_j(a)$ for each $a \in A$ and each $i$, $j \in I$. A potential game is a game with a potential function $\Phi: A \to \mathbb{R}$ satisfying for all $a_{-i} \in A_{-i}$ and all $a'_i$, $a''_i \in A_i$,

$$\Phi(a'_i, a_{-i}) - \Phi(a''_i, a_{-i}) = u_i(a'_i, a_{-i}) - u_i(a''_i, a_{-i}).$$

An ordinal potential game is game with a potential function $\Phi: A \to \mathbb{R}$ satisfying for all $a_{-i} \in A_{-i}$ and all $a'_i$, $a''_i \in A_i$,

$$\Phi(a'_i, a_{-i}) - \Phi(a''_i, a_{-i}) > 0 \Leftrightarrow u_i(a'_i, a_{-i}) - u_i(a''_i, a_{-i}) > 0.$$

Every identical interests game is a potential game and every potential game is an ordinal potential game.

## 2.3. The discrete replicator equation

Much of the background material for the population genetics models here is from [7] and [6].

Time is discrete and denoted by positive integers $t$. Let $f^t: \Delta^m \to \mathbb{R}^m$ be given for each time $t$, with $f_i^t: \Delta^m \to \mathbb{R}$ for each $1 \le i \le m$ being the standard coordinate projection of $f^t$.

The mean value function associated with $f^t$, denoted $\bar{f}^t$, maps $\theta \in \Delta^m$ to $\mathbb{R}$ by

$$\bar{f}^t(\theta) := \sum_{1 \le i \le m} \theta_i f_i^t(\theta).$$

The discrete replicator equation is then the recursive mapping from $\Delta^m$ to $\Delta^m$ defined by

$$\theta_i^{t+1} := \theta_i^t \frac{f_i^t(\theta)}{\bar{f}^t(\theta)}. \tag{2.1}$$

## 2.4. Alleles and genotypes

The model assumptions which will be maintained throughout are that populations are infinite (i.e. only proportions of genotypes and alleles in the population are of interest, not absolute numbers), that generations are discrete and non-overlapping, that selection occurs but not mutation or migration, and that stochastic genetic drift over time does not occur.

We assume that each gentoype is composed of $m$ genetic loci. Each locus $i$ is associated with a set of alleles $\mathcal{A}_i$ composed of $k+1$ alleles. A *genotype* is then formally a string $g = a_1 a_2 \ldots a_m$, such that $a_i \in \mathcal{A}_i$ for each $1 \le i \le m$. Denote the collection of all possible genotypes by $\Gamma$.

At each time $t$, there is an adult population $\Pi^t$ composed of individuals, each of which bears a genotype $g \in \Gamma$. The sub-population of individuals bearing genotype $g$ at time $t$ is denoted $\Pi_g^t$.

The adults in the population at time $t$ reproduce (asexually, haploid sexually, or diploid sexually, depending on the particular model being studied). After the adults in population $\Pi^t$ reproduce, an offspring population $\Omega^t$ comes into existence. The sub-population of individuals bearing genotype $g$ at time $t$ is denoted $\Omega_g^t$.

Denote by $d_g^t \in \Delta(\Gamma)$ the weight or relative proportion of genotype $g$ at time $t$ in the offspring population, i.e. the proportion of the set $\Omega_g^t$ in $\Omega^t$. At the beginning of period $t+1$, the adult population $\Pi^t$ dies, and as the individuals in $\Omega^t$ attain maturity they form the adult population $\Pi^{t+1}$.

A selection fitness value $w_g^t \in [0, 1]$ is associated with each genotype $g$ at each time $t$. This is interpreted as the probability that an offspring individual bearing genotype $g$ in population $\Omega^t$ will survive and attain reproductive maturity as an adult in population $\Pi^{t+1}$.

## 2.5. Asexual (clonal) reproduction model

In this model, at each time $t$, each individual in $\Pi_g^t$ produces $\zeta$ offspring (where $\zeta$ is a positive integer), each of whom bears the same genotype $g$ as its parent. The offspring thus produced in population $\Omega^t$ then mature into the adults in population $\Pi^{t+1}$, subject to selection as determined by $\{w_g^t\}_{g \in \Gamma}$.

The relevant state space of the dynamic is the simplex $\Delta(\Gamma)$. The *mean fitness* at time $t$ is

$$\bar{w}^t := \sum_{g \in \Gamma} d_g^t w_g^t. \tag{2.2}$$

In models in which $w_g^t$ is constant over time we may suppress the time denotation and simply write $w_g$, and hence $\bar{w}^t = \sum_{g \in \Gamma} d_g^t w_g$.

The dynamics of asexual reproduction are governed by the asexual replicator equation as the equation of motion,

$$d_g^{t+1} = d_g^t \frac{w_g^t}{\bar{w}^t}. \tag{2.3}$$

This follows the schema of equation (2.1), with the fitness $w_g^t$ in the role of $f_i$ and $\bar{w}$ corresponding to $\bar{f}$.

It will sometimes be convenient to express equation (2.3) generically as

$$d_g^+ = d_g \frac{w_g}{\bar{w}}, \tag{2.4}$$

suppressing reference to $t$ when its value is clear from context. In general, throughout this paper, we will write expressions such as $x^+$ in place of the longer $x^{t+1}$.

## 2.6. Haploid sexual reproduction model

This model will be central to much of the paper, hence we present its assumptions here in some detail. We suppose a monoecious sexually reproducing haploid population, with panmictic mating occuring in pairs. Initially, it will be supposed that there is no linkage between loci, i.e. each offspring at each locus bears the allele of one of the parents at the corresponding locus with equal probability. This assumption will be relaxed subsequently.

We will sometimes denote allele $j$ at locus $i$ by $a_{ij}$, where $1 \le i \le m$ and $1 \le j \le k + 1$, when convenient without confusion by context. When it is important to distinguish the $j$th allele in locus $i$ from the $j$th allele at locus $i'$, we will explicitly write $a_{ij_i}$. Let $C_{a_{ij_i}}$ denote the collection of all possible genotypes that contain $a_{ij_i}$ at the slot for locus $i$. Write $\Pi_{a_{ij_i}}^t := \bigcup_{g \in C_{a_{ij_i}}} \Pi_g^t$ and $\Omega_{a_{ij_i}}^t := \bigcup_{g \in C_{a_{ij_i}}} \Omega_g^t$.

Denote by $q_{ij_i}^t$ the *allelic frequency* of allele $a_{ij_i}$ at locus $i$ at time $t$ in population $\Omega^t$, i.e. $q_{ij_i}^t$ is the proportion of $\Omega_{a_{ij_i}}^t$ in $\Omega^t$. Call $q_i^t = \{q_{ij}^t\}_{1 \le j \le k+1}$ the *allelic frequency distribution* of locus $i$ at $t$; this is an element of a $k$-simplex, which we will denote $\Delta_i^k$. The relevant phase space for studying the evolutionary dynamic is then a polytope composed of an $m$-cross product of simplices:

$$\Theta := \Delta_1^k \times \cdots \times \Delta_m^k. \tag{2.5}$$

The topology for studying convergence is the product topology of the simplices regarded as manifolds.

Going from $\Delta(\Gamma)$ to $\Theta$ is always possible, since we defined $q_{ij_i}^t$ as the proportion of $\Omega_{a_{ij_i}}^t$ in $\Omega^t$ for each $a_{ij_i}$. Denote the mapping thus defined by $\rho : \Delta(\Gamma) \to \Theta$.

For $g = a_{1j_1} a_{2j_2} \ldots a_{mj_m}$ and an allelic frequency distribution $q^t \in \Theta$ denote

$$q_g^t = q_{1j_1}^t q_{2j_2}^t \ldots q_{mj_m}^t. \tag{2.6}$$

If $d_g^t = q_g^t$ for all $g$, a population is said to be in linkage equilibrium. When linkage equilibrium obtains, the inverse mapping $\rho^{-1} : \Theta \to \Delta(\Gamma)$ is well-defined by applying equation (2.6). When we make use of this inverse mapping, given $x \in \Theta$ we will write $[\rho^{-1}(x)]_g$ to stand for the $g$th component of $\rho^{-1}(x) \in \Delta(\Gamma)$.

The *marginal fitness* of allele $a_{ij}$ at time $t$ is defined as

$$w_{ij}^t := \sum_{g \in C_{a_{ij}}} w_g^t \frac{d_g^t}{\sum_{g' \in C_{a_{ij_i}}} d_{g'}^t}. \tag{2.7}$$

From the collection $\{w_{ij}^t\}_{1 \le j \le k+1}$, we furthermore can calculate the *mean payoff* for locus $i$, which is $\bar{w}_i^t := \sum_{j=1}^{k+1} q_{ij}^t w_{ij}^t$. But this yields nothing new, because $\bar{w}_i^t = \bar{w}^t$ of equation (2.2) for all loci $i$.

The dynamic in this model is the *haploid sexual replicator* which can be shown to be

$$q_{ij}^{t+1} = q_{ij}^t \frac{w_{ij}^t}{\bar{w}^t}, \tag{2.8}$$

and applies at every allele $j$ of every locus $i$. This clearly follows the schema of equation (2.1) with $w_i^t$ here in the role of $f^t$ in equation (2.1) and $\bar{w}_i^t$ as $\bar{f}^t$. As before, it will sometimes be convenient to express equation (2.8) generically as

$$q_{ij}^+ = q_{ij} \frac{w_{ij}}{\bar{w}}, \tag{2.9}$$

suppressing reference to $t$ when its value is clear from context.

The haploid sexual replicator dynamic maps points in $\Theta$ to points in $\Theta$, and hence also maps points in $\Delta_i^k$ to points in $\Delta_i^k$ under the projection from $\Theta$ to $\Delta_i^k$.

**Table 2.** Comparison of parallel notations used in this paper for game theoretic models and evolutionary reproduction models.

| game notation | evolution notation |
|---|---|
| action sets: $A = A_1, \ldots, A_m$ | alleles: $A = A_1, \ldots, A_m$ |
| pure action profile: $x = a_1, \ldots, a_m$ | genotype: $g = a_1 \ldots a_m$ |
| mixed strategy profile: $q$ | allelic frequency distribution: $q$ |
| payoff function: $u$ | fitness: $w$ |
| potential/objective function: $\phi$ | mean fitness: $\bar{w}$ |

### 2.6.1. Haploid reproduction as an identical interests game

In the fixed fitness case, the collection of fitness values $\{w_g\}_{g \in \Gamma}$ can be regarded as defining an identical interests game between the loci. Here, we essentially rewrite some of the previous sections in notation that is more familiar from the game theory literature, and unite the analysis of haploid dynamics with that of the dynamics of player strategies in repeated identical interests games when the players implement the haploid sexual replicator equation, equation (2.8), in updating their strategies.

From the fixed fitness haploid model with polytope $\Theta$ of alleles at the various loci, define an identical interest game $W_\Theta$ as follows. Each locus $i$ becomes a player $i$. The set of alleles of locus $i$ becomes the set of pure actions $A_i$ of player $i$. For each profile of pure actions $(a_1, \ldots, a_m) \in A_1 \times \cdots \times A_m$, the payoff $w_i(a_1, \ldots, a_m) = w(a_1, \ldots, a_m)$ to each player $i$ is identical and defined to be

$$w(a_1, \ldots, a_m) := w_g,$$

where $g = a_1 \ldots a_m$ is the genotype defined by $a_i \in \mathcal{A}_i$ for each $i$ and $w_g$ is fitness payoff to genotype $g$.

Here $q_i \in \Delta_i^k$, which previously denoted the distribution of alleles in locus $i$, is interpreted as a mixed strategy. The mean fitness $\bar{w}$ is interpreted as $\bar{w} = w(q_1, \ldots, q_m)$, the expected payoff (to each player in the game) when each player/locus $i$ plays mixed strategy $q_i$. The expected payoff/mean fitness $\bar{w}$ plays the role of the potential function in the identical interests game $W_\Theta$.

Every potential game (and hence every identical interests game) admits at least one pure strategy Nash equilibrium, namely the pure strategy profile yielding the highest potential payoff. The set of all pure Nash equilibria is the set of local maxima of the potential. Denote this set of pure Nash equilibria of $W_\Theta$ by $N_{W_\Theta}$.

Each $v \in N_{W_\Theta}$ is by definition a profile of alleles $(a_1, \ldots, a_m)$, one from each locus. Hence, it is naturally associated with a particular genotype that we will denote $g_v \in \Gamma$.

Note that if the set of mixed strategy profiles is restricted to a subset $\Theta' \subset \Theta$, a different identical interests game $W_{\Theta'}$ is induced. The set of pure Nash equilibria of $W_{\Theta'}$ may differ from the set of pure Nash equilibria of $W_\Theta$.

We may write $q_{i_{a_j}}$ as a synonym for $q_{ij}$ when $q_i$ is the mixed strategy of $i$. We can write $w(p; q_{-i})$ for the expected payoff when locus $i$ plays mixed strategy $p$ while all the other loci play mixed strategy $q_{-i}$. In a special case, this notation becomes $w(a_j; q_{-i})$, standing for the expected payoff when pure action/allele $a_j \in \mathcal{A}_i$ is chosen at locus $i$ while all the other loci play mixed strategy $q_{-i}$; this is none other than the game interpretation of the marginal fitness of allele $a_j \in \mathcal{A}_i$, which was above written as $w_{ij}$. Then for each $i$,

$$w(q_1, \ldots, q_m) = \sum_{a_\ell \in \mathcal{A}_i} q_{i_{a_\ell}} w(a_\ell; q_{-i}).$$

Table 2 presents in summary form comparisons between parallel notations used for the game theoretic models and the evolutionary reproduction models in this paper.

## 2.7. Diploid sexual reproduction model

### 2.7.1. One locus

In the single locus diploid model, with a set of alleles $\mathcal{A}$, one needs to keep track of pairs of alleles, $a_i a_j \in \mathcal{A}$, which constitute the genotypes. We suppose no position effects and hence do not distinguish between $a_i a_j$ and $a_j a_i$. Random mating is also assumed, hence Hardy–Weinberg ratios hold during the mating phase (with selection then moving the adult population away from the Hardy–Weinberg ratios).

Label the frequency of allele $a_i$ at time $t$ by $p_i^t$ and the frequency of genotype $a_i a_j$ by $P_{ij}^t = P_{ji}^t = p_i^t p_j^t$. Denote the fitness of genotype $a_i a_j$ by $W_{ij}^t = W_{ji}^t$, and the population mean fitness by

$$\bar{W}^t = \sum_{i,j} P_{ij}^t W_{ij}^t = \sum_i (p_i^t p_i^t W_{ii}^t + \sum_{j \neq i} p_i^t p_j^t W_{ij}^t). \tag{2.10}$$

Define the marginal fitness of allele $a_i$ as

$$W_i = \sum_j p_i W_{ij} = p_i W_{ii} + \sum_{j \neq i} p_j W_{ij}. \tag{2.11}$$

With all the preliminaries in place, the dynamic is once again defined by a straight-forward replicator as the equation of motion

$$p_i^+ = p_i \frac{W_i}{\bar{W}}, \tag{2.12}$$

for each allele.

### 2.7.2. Multiple loci

The multi-locus diploid model is complicated to describe; we omit most details and present only the minimal notation needed for our purposes here.

As before we suppose that there are $m$ loci with $k + 1$ alleles per locus. The state space is $\Delta_1^k \times \cdots \times \Delta_m^k$ and trajectories are elements $(p_1, \ldots, p_m) \in \Delta_1^k \times \cdots \times \Delta_m^k$.

Within each locus $i$, as in the single locus model, the alleles are between themselves playing at each time period a symmetric potential game with a fitness $W_{i_k i_l}^t$ assigned to each pairing $a_{i_k} a_{i_l}$, where $a_{i_k}, a_{i_l} \in A_i$. However, $W_{i_k i_l}^t$ is now a function not only of $a_{i_k} a_{i_l}$ but of the entire profile $p_{-i}$ of the allelic distributions of the other loci.

The standard analysis in the literature tracks the distribution of gametes (where each gamete is one possible haploid half of a diploid genotype). Each gamete $g$ can be assigned a marginal fitness $W_g$ as a function of the fitnesses of the pairings at each locus and the allelic frequency, and from this the mean fitness $\bar{W}$ of the population is calculated. Denoting the frequency of gamete $g$ by $r_g$, one can derive a recursion formula that is reminiscent of, but not identical to, the replicator equation

$$r_g^+ = r_g \frac{W_g}{\bar{W}} - D_g, \tag{2.13}$$

where $D_g$ is the linkage disequilibrium for $g$. The existence of the disequilibrium term $D_g$ means that the diploid multi-locus dynamic is not a replicator dynamic, making the analysis of this dynamic different from all the other models studied in this paper.

# 3. Multiplicative weights, potential games and virtual convergence

## 3.1. Regret minimization

The objective of many of the on-line learning algorithms developed in the literature in recent years is the attainment of regret minimization. Let $\mathcal{K} \subset \mathbb{R}^k$ be non-empty, bounded, compact and convex. At each iteration time $t$, algorithm $\mathcal{A}$ selects an element $x_t \in \mathcal{K}$, while a concave function $\ell_t : \mathcal{K} \to \mathbb{R}$ is revealed.

The goal of the algorithm is to minimize the average regret over any $n$ rounds, defined as

$$R_n(\mathcal{A}) := \sum_{t=1}^n \ell_t(x^*) - \sum_{t=1}^n \ell_t(x_t),$$

where $x^* \in \arg\max_{x \in \mathcal{K}} \sum_{t=1}^n \ell_t(x)$. In other words, the objective is to have minimal regret relative to having selected the best possible $x^* \in \mathcal{K}$ from the start and playing $x^*$ in a fixed manner at every time period.

An algorithm implements asymptotic regret minimization if its regret is sub-linear, i.e. $R_n(\mathcal{A}) = o(n)$ as $n \to \infty$. When this holds

$$\limsup_{n \to \infty} \frac{1}{n} \sum_{t=1}^n \ell_t(x^*) - \frac{1}{n} \sum_{t=1}^n \ell_t(x_t) \leq 0, \tag{3.1}$$

where $x^*$ is the element of $\mathcal{K}$ with the optimal average payoff.[2] In other words, the average regret converges to zero in the limit and the payoff of the algorithm approaches that of having selected the optimal in hindsight $x^*$ from the start and monotonically selecting only that point at every iteration.

## 3.2. Multiplicative weights update algorithm

The multiplicative weights update algorithm comes in two flavours: a polynomial and exponential version. In the polynomial version, $d \in \Delta^k$ is mapped to $d^+ \in \Delta^k$, conditional on receipt of a given tuple of real numbers $(\ell_1, \ldots, \ell_k)$, by

$$d_i^+ = \frac{d_i(1 + \eta\ell_i)}{\sum_j d_j(1 + \eta\ell_j)} \qquad (3.2)$$

for some $\eta > 0$. Dividing the numerator and denominator of equation (3.2) by $\eta$ changes nothing, hence equation (3.2) can be equivalently expressed as

$$d_i^+ = \frac{d_i(1/\eta + \ell_i)}{\sum_j d_j(1/\eta + \ell_j)}. \qquad (3.3)$$

In the special case in which $\eta \to \infty$, sometimes called the parameter-free version of the algorithm (cf. [11]), equation (3.3) becomes

$$d_i^+ = \frac{d_i\ell_i}{\sum_j d_j\ell_j} = d_i\frac{\ell_i}{\bar{\ell}}, \qquad (3.4)$$

which is exactly the replicator equation. (In this section, as in others, the term $d_i^+$ is written as meaning $d_i^{t+1}$).

The MWU dynamic determined by equation (3.2) can be given suggestive interpretations that relate it conceptually both to regret minimization and Nash equilibria. Consider that under equation (3.2) increases (respectively, decreases) the weight of $d_i$ if $\ell_i$ is greater (respectively, less) than the mean payoff $\sum_j d_j(1/\eta + \ell_j)$. This can be interpreted as the algorithm 'regretting' that it previously gave weight to the coordinates in $\Delta^k$ that yielded lower than average returns and correspondingly lowering their weights relative to the others in a retrospective attempt to lessen the regret.

For another interpretation of equation (3.2), note that $\ell_i$ is greater than the mean payoff then $\sum_j d_j(1/\eta + \ell_j)$ if and only if deviating from the mixed weighted $d \in \Delta^k$ to a distribution placing all the support on the $i$th coordinate would yield a higher payoff than the expectation of $d$ itself: an idea very close in spirit to the reasoning behind the Nash equilibrium solution concept.

In its exponential version, the multiplicative weights update algorithm, also known as the Hedge algorithm ([12])), maps $d \in \Delta^k$ to $d^+ \in \Delta^k$, conditional on receipt of a given tuple of real numbers $(\ell_1, \ldots, \ell_k)$, by

$$d_i^+ = \frac{d_i \cdot e^{\eta\ell_i}}{\sum_j d_j \cdot e^{\eta\ell_j}} \qquad (3.5)$$

for some $\eta > 0$.

The replicator equation can also be shown to be a special case of the exponential algorithm ([13]) as expressed in equation (2.1). The key is to register not the fitness payoffs at each time period but the logarithms of the fitnesses: given a fitness tuple $f = (f_1, \ldots, f_m)$, form the tuple $(\ell_1, \ldots, \ell_m)$ by setting $\ell_i = (1/\eta)\ln(f_i)$. Then apply equation (3.5):

$$d_i^+ = \frac{d_i \cdot e^{\eta\ell_i}}{\sum_j d_j \cdot e^{\eta\ell_j}} = \frac{d_i \cdot e^{\eta((1/\eta)\ln(f_i))}}{\sum_j d_j \cdot e^{\eta((1/\eta)\ln(f_j))}} = d_i\frac{f_i}{\bar{f}}.$$

It is well known in the literature that the multiplicative weights update algorithm attains regret minimization. In the genetic context studied here, this translates into attaining asymptotic average growth rates equal to that of having selected the optimal-in-hindsight genotype $g^*$ from the start and hypothetically running history again with a population consisting solely of $g^*$ at every time period.

---

[2]Strictly speaking we need to consider the lim sup in equation (3.1) because the limiting average payoff value might not be well defined.

Furthermore, since the haploid sexual reproductive strategy can be interpreted as an implementation of the replicator independently in each locus, the interpretation of the replicator as an instantiation of the multiplicative weights updating algorithm is applicable in several of the models in this paper, beyond the asexual model.

Several papers studying the applicability of multiplicative weights updating algorithms to evolutionary models have been published in recent years. A brief list of such papers includes [11,14–16].

## 3.3. Multiplicative weights and Baum–Eagon

It is instructive to compare the multiplicative weights updating algorithm, especially in its parameter-free version:

$$q_i^+ = q_i \frac{\ell_i}{\bar{\ell}},$$

and Baum–Eagon updating (as in equation (A 1))

$$q_{ij}^+ := q_{ij} \frac{\partial U/\partial q_{ij}}{\sum_s q_{is}(\partial U/\partial q_{is})}.$$

From the perspective of each $\Delta^i$, the Baum–Eagon updating is a special case of the multiplicative weights updating algorithm in which the payoff $\ell_{ij}$ is given as the partial derivative of a potential function $U$ with respect to $q_{ij}$. This perspective will have a significant role here, as many of the dynamics that will be studied benefit both from the monotonic potential increase afforded by the Baum–Eagon Theorem and the regret-minimization given by the multiplicative weights algorithm aspect.

## 3.4. Convergence in potential games

The content of the following theorem is technically equivalent to a theorem in [3] (see also [17]), which is expressed and proved there in the context of congestion games. We present it here with a full proof for two reasons: (a) an independent proof for potential games is of value; (b) the proof here can readily be understood in the context of reproductive strategies, such as haploid sexual reproduction, given the interpretation of such strategies as implementing the multiplicative weights updating algorithm, as described in §3.2, in the context of a potential game between loci, with alleles in the role of pure actions, as described in §2.6.1.

**Theorem 3.1.** *Suppose that each of a finite set of players playing a potential game implements the polynomial multiplicative weights update algorithm at discrete time periods to update his mixed strategy, starting from a mixed strategy of full support.*

*Then the strategy profile of the players will converge to a fixed point that is a Nash equilibrium.*

Theorem 3.1 is a stronger result than may appear at first glance, because there is no explicit coordinating element between the players that is assumed. To see why this may be surprising, consider the following extremely simple $2 \times 2$ game, which is an identical interests game (and hence a potential game):

|   | S | B |
|---|---|---|
| S | (2, 2) | (0, 0) |
| B | (0, 0) | (1, 1) |

One may interpret this as a coordination game between a couple, who wish to meet. If they are both at the symphony hall (action profile (S, S)) they each receive a payoff of 2; if they are both at the beach (action profile (B, B)) they each receive a payoff of 1; otherwise they fail to meet and receive zero payoff. Suppose that both players simultaneously implement a simple-minded best reply strategy, beginning at action profile (S, B). Then in the next time period, the action profile will be (B, S), followed by (S, B) etc. Lacking a coordinating element, no convergence to a fixed point is attained.

By contrast, theorem 3.1 does guarantee convergence under the multiplicative weights update algorithm, even though there is no coordination between the players and each player updates his or her mixed strategy from one time period to the next entirely independently of the other players. It is as if coordination is attained 'for free'. This result is attained by virtue of the Baum–Eagon Theorem, which underlies the proof of the theorem and guarantees that, despite the lack of coordination, a monotonic climb up the potential of the game ensues at each time period.

## 3.5. Virtual convergence

Let $\Theta := \Delta_1^k \times \cdots \times \Delta_m^k$ be a polytope, with $T : \Theta \to \Theta$ a transformation.

We will say that the dynamic defined by $T$ *converges polymorphically* if for each initial point $x \in \Theta$ there is a point $y \in \Theta$ such that $\lim_{n \to \infty} T^n(x) = y$. In the special case that for each $x$ the limit $y = \lim_{n \to \infty} T^n(x) = (q_1, \ldots, q_m)$ satisfies the condition that $q_i$ is a point distribution in $\Delta_i^k$ for each $1 \le i \le m$, the dynamic *converges monomorphically*.

Suppose now that a linear fitness function $\ell_t : \Theta \to \mathbb{R}$ is revealed for each time $t$. For an initial point $x \in \Theta$, denote $x_n := T^n(x)$, with $T^0(x) = x$. We will say that the dynamic defined by $T$ *virtually converges polymorphically* if for any sequence $\ell_1, \ell_2, \ldots$ of payoffs and any initial point $x \in \Theta$, there is a point $y^* \in \Theta$ such that

$$\left| \limsup_{n \to \infty} \frac{1}{n} \sum_{t=1}^n \ell_t(y^*) - \frac{1}{n} \sum_{t=1}^n \ell_t(x_t) \right| = 0.$$

In the special case that virtual convergence is to a $y^*$ that is a point distribution in $\Delta_i^k$ for each $1 \le i \le m$, we say that *virtual monomorphic convergence* obtains.

# 4. Asexual (clonal) reproduction

The dynamics of frequency independent asexual reproduction without mutation is perhaps the simplest of evolutionary dynamics—essentially 'bacteria in a Petri dish'. Despite the apparent simplicity, there is much to be said here that will also have implications for the analysis presented in later sections.

## 4.1. Fixed fitness

We suppose here a fixed fitness value $w_g$ for each genotype at each time period, generically with a genotype $g^* \in \Gamma$ whose fitness $w_{g^*}$ is maximal among the genotypes. There are several ways to analyse this; in the spirit of this paper, we may regard this dynamic as a single-player potential game. In this interpretation, there is one player whose mixed strategy at time $t$ is a probability measure $d^t \in \Delta(\Gamma)$ over the genotypes in $\Gamma$. The expected payoff is $\sum_{g \in \Gamma} d_g^t w_g$. Theorem 3.1 then implies convergence to a fixed point in $\Delta(\Gamma)$.

Alternatively, we may directly apply the Baum–Eagon Theorem. The dynamics are governed by the asexual replicator equation,

$$d_g^+ = d_g \frac{w_g}{\bar{w}}. \tag{4.1}$$

Since $\bar{w} = \sum_{g \in \Gamma} d_g w_g$, it follows that $\partial \bar{w} / \partial d_g = w_g$, hence equation (4.1) is an application of the Baum–Eagon transformation as expressed in equation (A 1). (As before, $d_i^+$ means $d_i^{t+1}$.)

Denote by $T_0 : \Delta(\Gamma) \to \Delta(\Gamma)$ the transformation that defines $d^+ = T_0(d)$ by mapping $d_g$ to $d_g^+$ for each $g$ according to equation (4.1). Since the population mean fitness is increasing monotonically, $\lim_{n \to \infty} T_0^n(d)$ for any starting distribution $d \in \Delta(\Gamma)$ converges to a point in $\Delta_1(\Gamma)$, i.e. a fixed point that is a point distribution, since the only fixed points of equation (4.1) are point distributions. All the weight is asymptotically on $1_{g^*}$, where $g^*$ is the genotype of maximal fitness.

This implies that the interior of the simplex $\Delta(\Gamma)$ forms a global exponentially stable basin of attraction. If, by contrast, the initial point lies within a strict subface $F \subset \Delta(\Gamma)$, then the convergence will again be to a monomorphic population whose genotype is the genotype of maximal fitness within $F$. This will clearly be sub-optimal if $g^* \notin F$.

## 4.2. Temporally varying fitness

The fixed fitness setting of asexual reproduction is the simplest evolutionary model, yielding perhaps the strongest result that can be expected, of monotonic and rapid fitness increase towards convergent fixation to the globally optimal genotype. This satisfactory result, however, may not necessarily obtain if fitnesses are no longer fixed in time.

In a temporally varying fitness model, we suppose that there is a collection of possible environments $\Omega$, such that each $\omega \in \Omega$ determines a fitness landscape such that each genotype $g$ is assigned a fitness

value $w_g(\omega)$ under $\omega$. At each time $t$ one environment $\omega$ from $\Omega$ is selected, with the payoff to the genotypes registered in accordance to the fitness landscape of that environment.

A simple hill climbing dynamic cannot be applicable here because there is a different 'hill' (i.e. fitness gradient derived from the environment) at each time period; the trajectory under the transformation $T_0$ will no longer be monotonically increasing in mean fitness. Despite this, the replicator algorithm does an excellent job at learning, even under conditions of temporally varying fitness. This can be seen in several ways.

Consider first a discrete i.i.d. model in which there is a probability measure $\mu$ over $\Omega$ determining the selection of the environment at each time period, repeated indefinitely. This determines for each genotype $g$ an expected fitness payoff under $\mu$. An optimal population will (generically) be composed of the genotype with maximal expected fitness payoff, and the replicator reliably identifies this genotype. More generally:

**Proposition 4.1.** *Let $(\Omega, \mathcal{B}, \mu)$ be a probability space over a collection $\Omega$ of environments. For each genotype $g \in \Gamma$, define a random variable $w_g(\omega) \in [0, 1]$, interpreted as the fitness of $g$ under environment $\omega \in \Omega$, from which the expected fitness is given as $E(w_g \mid \mu) = \int_\Omega w_g(\omega)\, d\mu(\omega)$.*

*Let $S : \Omega \to \Omega$ be a stationary and ergodic transformation defining a stochastic process for each $g$ by $w_g^t(\omega) = w_g(S^t(\omega))$. Then under the asexual replicator dynamic, with probability one the population asymptotically converges to a monomorphic population consisting of the genotype with maximal expected fitness.*

Proposition 4.1 indicates that when there is sufficient structure to the stochastic process of the varying environments, at least as expressed in stationary ergodicity (which include i.i.d. as a special case), the replicator dynamic will be able to extract the information inherent in the process to identify the optimal genotype and converge to that genotype, from any initial population state (that at least minimally includes the optimal genotype).

From here one can ask what happens when the stochastic process of varying environments can be any process at all. It is not difficult to conjure examples of temporally varying environments that do not admit convergence to a single genotype. For example, let

$$w_g^t = \begin{cases} e^{1/2} & \text{for } t \le 100 \bmod 200 \\ e^{1/3} & \text{for } t > 100 \bmod 200 \end{cases}$$

and

$$w_h^t = \begin{cases} e^{1/3} & \text{for } t \le 100 \bmod 200 \\ e^{1/2} & \text{for } t > 100 \bmod 200 \end{cases}$$

Then clearly both $\liminf (1/T) \sum_{t=1}^{T} \ln w_g^t < \limsup (1/T) \sum_{t=1}^{T} \ln w_h^t$ and $\liminf (1/T) \sum_{t=1}^{T} \ln w_h^t < \limsup (1/T) \sum_{t=1}^{T} \ln w_g^t$. When one genotype is strong the other is weak, each temporarily overtaking the other only to fall back later.

Nevertheless, it is possible to extend proposition 4.1 to much more general environments using the notion of *one-step-ahead expected log-fitness*. The one-step ahead expected log-fitness is the expected log-fitness of a generation conditional on the past generations.

**Definition 4.2.** *Let $(\Omega, \mathcal{B}, \mu)$ be a probability space over a collection $\Omega$ of environments and let $(\psi_t)_{t \ge 1}$ be a stochastic process of environments relative to $(\Omega, \mathcal{B}, \mu)$. For each genotype $g \in \Gamma$, define a process by $w_g^{\psi_t} = w_g(\psi_t)$, interpreted as the fitness of $g$ under environmental process $\psi_t$. Let $\rho_t^g = \ln w_g^{\psi_t}$ denote the log-fitness; assume that $\rho_t^g$ is always bounded.*

We will call $\hat{\rho}_t^g = (1/t) \sum_{s=1}^{t} E[\rho_{s+1}^g \mid \rho_s^g, \ldots, \rho_1^g]$ the average one-step-ahead expected log-fitness of $g$ at $t$.

**Definition 4.3.** *A genotype $g$ is asymptotically one-step-ahead superior on average if $\liminf_{t \to \infty} \hat{\rho}_t^g > \limsup_{t \to \infty} \hat{\rho}_t^h$ with probability one for all genotypes $h \in \Gamma$ with $h \ne g$.*

**Theorem 4.4.** *If a genotype $g \in \Gamma$ is asymptotically one-step-ahead superior on average then, under the asexual replicator dynamic, with probability one the population asymptotically converges to a monomorphic population consisting of the genotype $g$.*

It is worthwhile noting here that in the case of an ergodic environment, $\liminf_{t \to \infty} \hat{\rho}_t^g$ and $\limsup_{t \to \infty} \hat{\rho}_t^g$ are one and the same and equal to the constant $E[\ln w_g]$ almost surely. Thus the sufficient condition established in theorem 4.4, namely asymptotic one-step-ahead superiority, is reduced to $E[\ln w_g] > E[\ln w_h]$.

The statement of theorem 4.4 supposes that genotype $g$ is asymptotically one-step-ahead superior on average with probability one with respect to all environments. Suppose instead that a genotype $g$ is asymptotically one-step-ahead superior on average only with respect to a subset $A_g^\infty$ of the collection of environments. Then we obtain the following corollary.

**Corollary 4.5.** *If* $\Pr(A_g^\infty) > 0$, *where* $A_g^\infty$ *is the set of environments in which genotype g is asymptotically one-step-ahead superior on average, then under the asexual replicator dynamic, with probability one in* $A_g^\infty$ *the population asymptotically converges to a monomorphic population consisting of the genotype g.*

Algorithms such as the multiplicative weights and mirror ascent algorithms have been developed in the computer science literature in recent years for the sake of optimization under conditions of no statistical structure. The replicator dynamic, it turns out, exploits the results afforded by these algorithms.

**Theorem 4.6.** *Under the replicator dynamic, for any arbitrary temporally varying fitness there is an optimal-in-hindsight genotype* $g^*$ *such that for any initial point in the interior of the simplex, asexual reproduction virtually converges monmorphically to* $g^*$.

In summary, we interpret the results of this section from a learning perspective: the objective is to learn which genotype is best fit for the environment process, via the algorithmic tool of the replicator.

When the environment is fixed, the replicator homes in on the objectively fittest genotype. When the environment process is sufficiently structured, as in a stationary ergodic process, the replicator makes use of time averaging to identify a winning genotype. Failing that, in the worst case in which there is insufficient structure for predictive learning, the replicator still manages to extract information, by application of regret minimization via the multiplicative weights updating algorithm; virtual convergence occurs in the sense that one can imagine a population which from the start consisted of only the optimal-in-hindsight genotype and attaining the same asymptotic average growth rate as actually attained.

# 5. Haploid sexual reproduction

## 5.1. Fixed fitness

In this section, the population will be presumed to reproduce via haploid sexual reproduction under a fitness landscape $\{w_g\}_{g \in \Gamma}$ that is fixed throughout time.

### 5.1.1. Under linkage equilibrium

Under linkage equilibrium, in population $\Omega_g^t$ the equation $d_g^t = q_{1j_1}^t q_{2j_2}^t \ldots q_{mj_m}^t$ holds for each genotype $g = a_{1j_1} a_{2j_2} \ldots a_{mj_m}$. As $\Omega_g^t$ matures into $\Pi_g^{t+1}$, selection applies such that linkage equilibrium does *not* hold for $\Pi_g^{t+1}$; however, by assumption random mating between the reproducing adults in $\Pi_g^{t+1}$ immediately restores linkage equilibrium in the next offspring generation $\Omega_g^{t+1}$.

One advantage of working with an assumption of linkage equilibrium is that we may identify in a bijective manner a point in the allelic frequency space $\Theta$ and a corresponding point in the genotypic frequency space $\Delta(\Gamma)$. We shall freely do so in this section as follows.

Recalling the haploid sexual replicator,

$$q_{ij_i}^+ = q_{ij_i} \frac{w_{ij_i}}{\bar{w}}, \tag{5.1}$$

define $\tau : \Theta \to \Theta$ to be the transformation given by the mapping of $q_{ij}$ to $q_{ij}^+$ for each locus $i$ and allele $j$ in $i$. Exploiting the linkage equilibrium assumption, define a transformation $T_1 : \Delta(\Gamma) \to \Delta(\Gamma)$ by

$$T_1(d) = \rho^{-1} \circ \tau \circ \rho(d). \tag{5.2}$$

Abusing terminology, we will call both $\tau$ and $T_1$ haploid sexual replicator transformations. This enables us to analyse the dynamics equally well under either $T_1$ or $\tau$; both define discrete dynamical systems determining trajectory paths in $\Delta(\Gamma)$ or in $\Theta$, respectively.

The Baum–Eagon inequality applies here with respect to the haploid reproduction dynamic, hence it follows immediately that mean fitness increases monotonically until a fixed point of the dynamic is attained.[3] The domain is the polytope $\Theta$ as defined in equation (2.5).

**Lemma 5.1.** *The haploid sexual replicator transformation (under linkage equilibrium and without genetic linkage between loci) satisfies the Baum–Eagon inequality, with mean fitness* $\bar{w}$ *as a Lyapunov function.*

It follows that the population will asymptotically converge to a fixed point of the dynamics defined by the transformation $T_1$ along paths of monotonically increasing mean fitness. In fact, a stronger

---

[3]This result is actually mentioned, without a detailed proof, all the way back in the original paper by Baum & Eagon [4].

statement can be made: assuming that the Nash equilibria of the associated game are isolated (which is true generically), convergence will not only be to a fixed point, it will be to a Nash equilibrium (which is a strict subset of the set of fixed points).[4]

**Theorem 5.2.** *Under haploid recombinative sexual reproduction (under linkage equilibrium and without genetic linkage between loci), trajectories almost always increase mean fitness monotonically.*

*Beginning from almost any interior point of $\Delta(\Gamma)$ the haploid sexual replicator dynamic converges asymptotically to a monomorphic population in which each individual bears a genotype $g_\nu$ from the set $N_{W_\Theta}$ of pure Nash equilibria of the associated potential game $W_\Theta$.*

**Corollary 5.3.** *If the initial point of the allelic frequency of the population lies in any face $\Theta'$ of $\Theta$ then the dynamic converges asymptotically to a monomorphic population consisting of genotypes from the set of pure Nash equilibria of the associated potential game $W'_\Theta$.*

There are immediate interesting implications of theorem 5.2. One of these is that $\Delta(\Gamma)$ is entirely partitioned into asymptotically stable basins of attraction (deterministically in this model).

**Theorem 5.4.** *For each pure Nash equilibrium $\nu \in N_{W_\Theta}$, there exists $B_\nu \subset \Delta(\Gamma)$ containing $1_{g_\nu} \in \Delta_1(\Gamma)$ such that starting from any initial point in $B_\nu$ the population under the dynamic will converge to a monomorphic population consisting solely of genotype $g_\nu$, i.e. $T_1^n(x) \longrightarrow 1_{g_\nu}$ for every $x \in B_\nu$. Apart from separatrices between these basins of attraction, which are of negligible measure, the sets in the collection $\{B_\nu\}$ form a partition of $\Delta(\Gamma)$.*

Even more than that can be said here. By theorem 6 of [2], any transformation of the form defined in equation (A 1) increases $U$-homotopically, from which it follows that the haploid sexual replicator transformation $T_1 : \Delta(\Gamma) \to \Delta(\Gamma)$ increases $\bar{w}$-homotopically.

**Proposition 5.5.** *Let $S_t(x) = tT_1(x) + (1 - t)x$. For each pure Nash equilibrium $\nu \in N_{W_\Theta}$, there exists a neighbourhood $H_\nu \subset \Delta(\Gamma)$ of $1_{g_\nu}$ such that $S_t(H_\nu) \subset H_\nu$ for $0 < t \leq 1$, and for every $x \in H_\nu$, $T_1^n(x) \longrightarrow 1_{g_\nu}$. Furthermore, each $H_\nu$ has the homotopy type of a disk.*

The significance of the 'basin of homotopic attraction' $H_\nu$ of proposition 5.5 is that not only does every point $x \in H_\nu$ converge to $g_\nu$ under the dynamic, also a small perturbation of around $x$ preserves this property. By contrast, around any pure strategy point that is neither a local maximum or a local minimum there are points such that a small perturbation can lead to asymptotic convergence to different fixed points.

Finally:

**Proposition 5.6.** *For each pure Nash equilibrium $\nu \in N_{W_\Theta}$, there exists a neighbourhood $E_\nu \subset \Delta(\Gamma)$ of $1_{g_\nu}$ that is an exponentially stable basin of attraction.*

Within the exponentially stable basin of attraction around a Nash equilibrium, the haploid sexual replicator dynamics resembles the asexual replicator dynamics, with exponential convergence to an equilibrium point.

The containment relations are $1_{g_\nu} \in E_\nu \subseteq H_\nu \subseteq B_\nu$. This implies that an observer following a trajectory starting in $B_\nu$ far from $g_\nu$ will likely initially see a slow and moderate increase in mean fitness, with broad polymorphism, for a long time, but once the trajectory enters $E_\nu$ suddenly an extremely fast rise in mean fitness will be registered along with rapid convergence to a monomorphic population.

### 5.1.2. Under genetic linkage between loci

Genetic linkage in this section means physical linkage between loci: in the context of haploid reproduction, this means that an offspring zygote might inherit a pair (or more) of alleles from one of the parents as a package, in contrast to an assumption of independent inheritance with probability 0.5 from each parent at each locus.

The term linkage disequilibrium refers to a population genotype distribution that does not equal the cross product of the marginal distribution as reflected in the allelic distribution. We note that in the game theoretic terms that we have been applying throughout to the study of genetic reproduction, linkage equilibrium corresponds to independent strategy selection of each player, while linkage disequilibrium corresponds to dependencies in the selection of strategies.

---

[4]This important point, that convergence is to a Nash equilibrium and not only to the set of fixed points of the dynamic (which actually includes any distribution with support on one genotype), seems to be missing from several accounts in the literature applying the Baum–Eagon inequality in the evolutionary setting. See for example [5, Appendix B], [6, p. 47] and [7, p. 34].

We initially present an analysis of the haploid sexual replicator under genetic linkage between loci in the case of two loci, for clarity of exposition.

Let $r \in [0, 1]$ be the recombination rate. Suppose one starts with a point $d \in \Delta(\Gamma)$ representing the population distribution. Project $d$ to $\Theta$ via $\theta = \rho(d)$. Under the asexual replicator $d$ is mapped to $T_0(d)$, and under the haploid replicator $\theta$ is mapped to $\tau(\theta)$. Then the replicator equation under recombination rate $r$ is

$$d^+ = r[\rho^{-1}(\tau(\theta))] + (1 - r)\alpha(d)$$

or, using the transformation $T_1$ defined in equation (5.2),

$$d^+ = rT_1(d) + (1 - r)T_0(d). \tag{5.3}$$

We may denote by $T_r$ the transformation defined by equation (5.3) which is consistent with our labelling of $T_1$ and $T_0$. When $r \neq 1$, genetic linkage between the loci occurs.

The recombination rate $r$ is intended to describe a situation in which each offspring is produced by sexual recombination with probability $r$ and is produced by asexual cloning with probability $1 - r$. This results in an offspring population, such that within that population, a weight $r$ of the offspring is descended from a sexual reproduction event and weight $1 - r$ is descended from an asexual reproduction event.

We may instead consider the following situation, which is mathematically equivalent and more convenient for our purposes: create two separate copies $\Pi_0$ and $\Pi_1$ of the reproducing population $\Pi$, maintaining the genotype frequencies of the original population in each copy, with relative population size proportions $|\Pi_1|/|\Pi_0| = r/(1 - r)$. Let $\Pi_0$ reproduce asexually to produce offspring population $\Omega_0$ and $\Pi_1$ reproduce haploid sexually to produce offspring population $\Omega_1$, finally combining them into $\Omega = \Omega_0 \cup \Omega_1$ and regarding the genotypic frequency of $\Omega$.

Slightly more generally, select fraction $r$ of the population at random to reproduce by the haploid sexual transformation, with the remaining $1 - r$ of the population reproducing by the asexual transformation. All of these alternatives result in an offspring population with weight $r$ descending from a sexual reproduction event and weight $1 - r$ descending from an asexual reproduction event, which is what is relevant.

**Proposition 5.7.** *In a population starting at an initial point in linkage equilibrium, under two-locus haploid recombinative sexual reproduction with recombination rate r, trajectories always increase mean fitness monotonically.*

*Beginning from any such interior point of $\Delta(\Gamma)$ the haploid sexual replicator dynamic converges asymptotically to a monomorphic population in which each individual bears a genotype $g_v$ from the set $N_{W_\Theta}$ of pure Nash equilibria of the associated potential game $W_\Theta$.*

In greater generality, suppose that there are $m$ loci. Let $\lambda$ be a partition of $\{1, \ldots, m\}$ into $\ell \leq m$ partition elements. An individual will be of $\lambda$-type if, when reproducing, the genes of that individual undergo physical genetic linkage according to $\lambda$. In other words, if two $\lambda$-type individuals $I_1$ and $I_2$ mate and produce an offspring $O$, then for each partition element $\bar{\lambda}$ of $\lambda$, all the alleles in the loci included in $\bar{\lambda}$ in the genotype of $O$ will be identical to either the alleles of $\bar{\lambda}$ in the genotype of $I_1$ or the alleles of $\bar{\lambda}$ in the genotype of $I_2$, with equal probability. If the entire population reproduces in this way, denote the resulting transformation from $\Delta(\Gamma)$ to $\Delta(\Gamma)$ by $T_\lambda$.

If $\lambda$ is the coarsest partition, consisting of only one partition element, this describes asexual reproduction. For any other partition, $\lambda$-type reproduction with $1 < \ell \leq m$ partition elements reduces to haploid sexual reproduction: simply regard the $\ell$ partition elements as $\ell$ independent loci. If $\lambda$ is the finest partition, in which each locus is its own partition element, this describes haploid sexual reproduction under independence of inheritance at each locus.

Let $\Lambda$ be the set of all partitions of $\{1, \ldots, m\}$. For each $\lambda \in \Lambda$ let $r_\lambda \in [0, 1]$, such that $\sum_{\lambda \in \Lambda} r_\lambda = 1$. Interpret $r_\lambda$ as the probability that an offspring is produced by physical genetic linkage in accordance with partition $\lambda$. Mathematically, this is equivalent to selecting at random at each generation, for each $\lambda \in \Lambda$, a fraction $r_\lambda$ of the population which reproduces by $\lambda$-type reproduction.

The resulting $\{r_\lambda\}_{\lambda \in \Lambda}$-tuple replicator equation is

$$d^+ = \sum_{\lambda \in \Lambda} r_\lambda T_\lambda(d). \tag{5.4}$$

**Theorem 5.8.** *In a population starting at an initial point in linkage equilibrium, under m-locus haploid recombinative sexual reproduction with recombination tuple $\{r_\lambda\}_{\lambda \in \Lambda}$, trajectories always increase mean fitness monotonically.*

**Table 3.** Two examples of fitness matrices for haploid reproduction with two loci and three alleles per locus.

| | $a_{21}$ | $a_{22}$ | $a_{23}$ |
|---|---|---|---|
| $a_{11}$ | $w_{a_{11},a_{21}} = 0.40$ | $w_{a_{11},a_{22}} = 0.60$ | $w_{a_{11},a_{23}} = 0.80$ |
| $a_{12}$ | $w_{a_{12},a_{21}} = 0.48$ | $w_{a_{12},a_{22}} = 0.55$ | $w_{a_{12},a_{23}} = 0.75$ |
| $a_{13}$ | $w_{a_{13},a_{21}} = 0.20$ | $w_{a_{13},a_{22}} = 0.51$ | $w_{a_{13},a_{23}} = 0.70$ |
| | $a_{21}$ | $a_{22}$ | $a_{23}$ |
| $a_{11}$ | $w_{a_{11},a_{21}} = 0.40$ | $w_{a_{11},a_{22}} = 0.48$ | $w_{a_{11},a_{23}} = 0.20$ |
| $a_{12}$ | $w_{a_{12},a_{21}} = 0.60$ | $w_{a_{12},a_{22}} = 0.55$ | $w_{a_{12},a_{23}} = 0.51$ |
| $a_{13}$ | $w_{a_{13},a_{21}} = 0.80$ | $w_{a_{13},a_{22}} = 0.75$ | $w_{a_{13},a_{23}} = 0.70$ |

*Beginning from any such interior point of $\Delta(\Gamma)$ the haploid sexual replicator dynamic converges asymptotically to a monomorphic population in which each individual bears a genotype $g_\nu$ from the set $N_{W_\Theta}$ of pure Nash equilibria of the associated potential game $W_\Theta$.*

The conclusion is that whether or not there is genetic linkage, in haploid sexual reproduction mean fitness increases monotonically and the population always converges to a monomorphic population corresponding to a pure Nash equilibrium of the potential game (this statement also holds true for asexual reproduction, since the equilibrium of maximal mean fitness is itself a pure Nash equilibrium of the potential game). Results similar to those in theorem 5.4 and proposition 5.5 also attain whether or not there is genetic linkage, with sensitivity to initial conditions as before.

However, although under any genetic linkage structure convergence to some pure Nash equilibrium occurs, the probability of converging to any particular pure Nash equilibrium, starting from the same initial allelic distribution, differs from one linkage structure to another. The same initial point can converge to different equilibria points depending on the linkage structure (as can be seen for example in the extreme case of no recombination, under which convergence will always be to the asexual globally optimal fitness equilibrium.)

### 5.1.3. External and internal environments

Let $\mathcal{E}$ be a collection of possible environments. Each $e \in \mathcal{E}$ determines a fitness landscape in the sense that $e$ is identified with an identical interests game $W_e$ played by the loci, relative to a fixed allelic frequency space $\Theta$.

In this section, we will make a distinction between what we term the 'external environment' and the 'internal environment'. The external environment is the realization $e^1, e^2, \ldots, e^t, \ldots$ and the corresponding $W_{e^1}, W_{e^2}, \ldots, W_{e^t}, \ldots$. Parallel to this, we may take the perspective of any particular locus $i$. From this perspective, the alleles in locus $i$ are among themselves implementing an asexual replicator dynamic as follows. Let $(q_1^t, \ldots, q_m^t)$ denote the profile of allelic frequencies over time. At time $t$, the identical interests game is $W_{e^t}$, and we may write the time $t$ growth rate as the stage $t$ game payoff $w_{e^t}(q_1, \ldots, q_m)$. Let $q_{-i}^t = (q_1^t, \ldots, q_{i-1}^t, q_{i+1}^t, \ldots, q_m^t)$ denote the profile of the $m-1$ loci apart from $i$. Call the sequence $(q_{-i}^1, q_{-i}^2, \ldots)$ the *internal environment* from the perspective of locus $i$.

Define the *total environment* from the perspective of locus $i$ at time $t$ to be $w_{e^t}(\cdot; q_{-i}^t)$, meaning that each choice of $q_i \in \Delta(A_i)$ yields the payoff $w_{e^t}(q_i; q_{-i})$. In this way, we may reduce the dynamic of each locus to the asexual replicator, with the alleles in locus $i$ implementing the asexual dynamic with respect to the total environment from their perspective.

### 5.1.4. Virtual convergence

Table 3 presents examples of two fitness matrices with two loci and three alleles per locus. One may interpret these as representing a population that is exposed to two possible environments, one per matrix, where the top is interpreted as a 'rainy year' environment and the bottom one is a 'drought year' environment.

The top matrix has one pure Nash equilibrium, at $(a_{11}, a_{23})$, and the bottom matrix similarly one at $(a_{13}, a_{21})$. It is possible, however, for an environment realization to create a trajectory that almost always remains extremely close to points that are not Nash equilibria—even though from the perspective of both matrices there is a repulsion from those points. For example, there can be a trajectory that begins close to $(a_{13}, a_{21})$ and

remains there under an alternating realization, rainy one year and drought the next. This is because although at each time period the single period dynamic pushes away from $(a_{13}, a_{21})$, the directions of push away from that point are nearly opposite, hence the trajectory never wanders far.

Alternatively, it is possible to imagine environment realizations in which very long stretches of one environment bring the population very nearly to convergence to $(a_{11}, a_{23})$, followed by equally long stretches subsequently driving the population very nearly to convergence to $(a_{13}, a_{21})$, repeated periodically in such a way that there is no candidate even for 'near convergence'. These simple examples indicate that the behaviour under time varying fitness can be complex and highly dependent on initial conditions and random realizations. Nevertheless, we can state the following theorem on virtual convergence. The intuition behind it is that even when the external and internal environments change upredictably, from the perspective of each individual locus the internal alleles are implementing the simple replicator equation, hence each locus experiences virtual convergence.

**Theorem 5.9.** *The haploid sexually reproducing dynamic virtually converges monomorphically under any environment realization.*

Note that although the result of theorem 5.9 guarantees virtual monomorphic convergence under every environment realization, different realizations can lead to different virtual growth rates. If the external environment process is sufficiently regular, however, then every realization will lead to the same virtual convergence, even though the corresponding internal environment process may not follow parallel regularity.

**Proposition 5.10.** *If the external environment follows a stationary and ergodic stochastic process then every environment realization leads to the same virtual monomorphic growth rate.*

# 6. Diploid sexual reproduction

## 6.1. Single locus model

### 6.1.1. Fixed fitness

The diploid sexual reproduction in the single locus model is extremely similar to the haploid two locus model, which enables many of the results from haploid sexual reproduction to be carried over almost entirely (and arguably relatively simply, since the parallel is to two loci and not $m$ loci) but for one very significant difference: where in the haploid two locus model fitness is represented by a potential matrix with a separate set of alleles for the row player and the column player (corresponding to different alleles in the different loci), in the single locus diploid model the same alleles appear as both row players and column players in the matrix.

The state space is the allelic frequency space $\Theta = \Delta(A)$, with the trajectories in $\Delta(A)$ recursively following the replicator equation. The fitness landscape[5] determined by fitness $W_{ij}$ corresponding to gamete $a_i a_j$, denoted here as before by $W_\Theta$ is a symmetric matrix.

Dynamics with respect to symmetric matrices have long been studied in the literature of evolutionary game theory. The parallels are clear: both the diploid single locus and the population dynamic cases can be thought of as a single player game, in which the player selects a mixed strategy (e.g. the ratio of hawks to doves in the population, or the ratio of allele A to allele B), receives an expected payoff, and in the next time period updates the mixed strategy in accordance with a replicator equation. In the evolutionary game theory literature, it is well known that such a dynamic leads to convergence to evolutionarily stable Nash equilibria, which may be either pure or mixed Nash equilibria.

In comparison with the haploid sexual model, the significant new element is, of course, the possibility of convergence to mixed Nash equilibria. One possible evolutionary advantage of maintaining mixed equilibria versus convergence to pure equilibria is that mixed equilibria may be similar to constantly re-balanced portfolios in investment theory; it is well known that re-balanced portfolios (analogous to mixed equilibria) can significantly outperform single stock portfolios (analogous to pure equilibria).

There is one major difference between the dynamics of the diploid sexual reproduction and the population dynamics of a typical evolutionary game theory model. In the fixed fitness/matrix setting, evolutionary game theory models can frequently exhibit cyclic or chaotic trajectories that never

---

[5]We assume here that there are no position effects.

converge. By contrast, diploid single locus dynamic trajectories are generically monotonically increasing in mean fitness and converge, to either a stable monomorphic population (pure Nash equilibrium) or a stable polymorphic population (mixed Nash equilibrium).

This is because the diploid single locus dynamic follows the Baum–Eagon inequality, montonically increasing in mean fitness. Theorem 3.1 is not directly applicable here, because of the nonlinearity in the payoffs to the players with respect to the payoff matrix. One can instead show directly from the equations of motion that the Baum–Eagon inequality holds. This result is known in the literature (e.g. [6]); we reproduce it here for completeness.

Recall that by equation (2.11), $W_i = \sum_j p_j W_{ij}$ and that by equation (2.10), $\bar{W} = \sum_i (p_i^2 W_{ii} + \sum_{j \neq i} p_i p_j W_{ij})$. Calculating the partial derivative, $\partial \bar{W} / \partial p_i = 2 p_i W_{ii} + \sum_{j \neq i} 2 p_j W_{ij} = 2(p_i W_{ii} + \sum_{j \neq i} p_j W_{ij})$ (taking into account the fact that $i$ will appear both in $W_{ij}$ and $W_{ji}$ for each $j$).

It follows that $\sum_j p_i (\partial W_i / \partial p_i) = 2(\sum_i (p_i^2 W_{ii} + \sum_{j \neq i} p_i^t p_j^t W_{ij})) = 2\bar{W}$. Hence

$$p_i \frac{\partial \bar{W}/\partial p_i}{\sum_j p_j (\partial \bar{W}/\partial p_j)} = p_i \frac{2(p_i W_{ii} + \sum_{j \neq i} p_j W_{ij})}{2\bar{W}}$$

$$= p_i \frac{2W_i}{2\bar{W}} = p_i \frac{W_i}{\bar{W}}$$

$$= p_i^+,$$

with the last equality following equation (2.12). Hence the Baum–Eagon Theorem applies to the dynamic in $\Delta(A)$ and one concludes that in the single locus diploid sexual reproduction dynamic fitness monotonically increases until a local maximum is attained.

The upshot is that, apart from the possible convergence to stable polymorphism when mixed strategies are the end result of the dynamic, the diploid single locus model parallels the haploid two-locus model in the crucial aspects of monotonic Baum–Eagon mean fitness increase while following a replicator recursion. This enables us to adapt many of the results from the haploid analysis to the diploid model.

**Theorem 6.1.** *In the diploid single-locus model, for each ESS equilibrium $v$, there exists $B_v \subset \Delta(\Gamma)$ containing $1_{g_v}$ such that starting from any initial point in $B_v$ the population under the dynamic will converge to a monomorphic population consisting solely of genotype $g_v$, i.e. $T_1^n(x) \longrightarrow 1_{g_v}$ for every $x \in B_v$. Apart from separatrices between these basins of attraction, which are of negligible measure, the sets in the collection $\{B_v\}$ form a partition of $\Delta(\Gamma)$.*

**Proposition 6.2.** *Let $S_t(x) = tT_1(x) + (1-t)x$. In the diploid single-locus model, for each ESS equilibrium $v$, there exists a neighbourhood $H_v \subset \Delta(\Gamma)$ of $1_{g_v}$ such that $S_t(H_v) \subset H_v$ for $0 < t \leq 1$, and for every $x \in H_v$, $T_1^n(x) \longrightarrow g_v$. Furthermore, each $H_v$ has the homotopy type of a disk.*

### 6.1.2. Temporally variable fitness

The time varying fitness results of the haploid two-locus model similarly carry over to the diploid single-locus model.

**Theorem 6.3.** *The single-locus diploid sexually reproducing dynamic under varying fitness environments virtually converges polymorphically under any environment realization.*

## 6.2. Multiple locus model

The diploid multi-locus model under linkage equilibrium re-capitulates the diploid single locus model at every locus. Hence the results of the previous section apply without changes.

Many of the tools of the previous sections fail to apply in the diploid multi-locus model under linkage disequilibrium. The main equation of motion, equation (2.13), is similar to but not quite a replicator equation. More to the point, the presence of the disequilibrium term causes the dynamic to fail to conform to the Baum–Eagon conditions. Hence the Baum–Eagon theorem, even under fixed fitness, cannot be used to conclude that monotonic fitness increase occurs; indeed, it has long been known that there are examples of fitness landscapes in which diploid populations can exhibit reductions in mean fitness over stretches of time and even periodically cycling trajectories. This leaves no possibility for a general theorem on either monomorphic or polymorphic convergence.

However, virtual convergence (possibly polymorphic), which does not depend on monotonic fitness increase, does obtain here under both fixed and temporally varying fitness. The proof is essentially the same as the proof of haploid virtual convergence (theorem 5.9) under temporally varying fitness. As before, we suppose a choice of environment realization $e^1, e^2, \ldots$ from a set of possible environments,

and distinguish between the external environment, represented by such a realization, and the internal environment perceived by a locus $i$, which is the allelic frequencies of the other loci at any time $t$.

**Theorem 6.4.** *The multi-locus diploid sexually reproducing dynamic virtually converges polymorphically under any environment realization.*

# 7. Conclusion

There are two main themes in what has been shown in this paper: (1) when environments are fixed, the monotonic climb of payoff values towards pure Nash equilibrium convergence in potential games under independent application of the MWU algorithm is exploited by evolutionary reproduction in several of its versions: asexual, haploid sexual, and single-locus diploid; (2) when environments vary, these evolutionary reproduction algorithms asymptotically approach what we have termed virtual convergence, under which they attain the best outcome that they could have attained in hindsight by implementing a pure strategy (furthermore, in the haploid sexual case all these obtain even under genetic linkage).

These results shed light on the mechanisms used by evolutionary processes to attain near optimal growth rates, providing a unified mathematical framework for understanding some known results on fixed environment convergence in the asexual, haploid sexual and single-locus diploid sexual reproduction models while also presenting novel results for both fixed and varying environment population genetics models.

We have also proved several results that provide details regarding the topological structure of the trajectories and basins of attraction of the haploid sexual reproduction model. We leave to future research the possible use of these insights into the role of mutations.

Data accessibility. We have supplied the code for the simulation generating figure 1.

Authors' contributions. All the authors contributed to the design of the model and the discussion of the results. O.E. and Z.H. conceived and proved the mathematical results. I.N. contributed to the simulations illustrating the mathematical results (see figure 1).

Competing interests. We declare we have no competing interests.

Funding. This work was partially supported by Israel Science Foundation grant no. 1626/18.

Acknowledgements. The authors thank Daniel B. Weissman for suggesting corrections to an earlier version of this paper, as well as two anonymous reviewers.

# Appendix A. Baum–Eagon dynamics

We make use extensive use here of the Baum–Eagon inequality (originally developed for the study of hidden Markov models by use of the Baum–Welch algorithm). The concepts and results in this section are from [4] and [2]. A brief exposition on the Baum–Eagon inequality with applications to population genetics appears in [6].

## A.1. Baum–Eagon inequality

Let $\Theta$ be a polytope given by a cross product of simplices,[6] i.e. $\Theta := \Delta_1^k \times \cdots \times \Delta_m^k$. Denote the $j$th element in the $i$th simplex by $x_{ij}$.

Let $U(x_{ij})$ be a real-valued polynomial function with non-negative coefficients over the variables $\{x_{ij}\}_{i,j}$. Let $\mathbf{x}$ be a point in the domain $\Theta$. Let $T(\mathbf{x})$ denote the point of $\Theta$ whose $i$, $j$th coordinate is given by

$$T(\mathbf{x})_{ij} := x_{ij} \frac{\partial U/\partial x_{ij}|_{\mathbf{x}}}{\sum_s x_{is}(\partial U/\partial x_{is})|_{\mathbf{x}}}, \tag{A 1}$$

where the denominator is a normalizing element.

Then $U(T(\mathbf{x})) > U(\mathbf{x})$ unless $T(\mathbf{x}) = \mathbf{x}$. ∎

It is possible to give the Baum–Eagon inequality an interesting gradient interpretation. Fix $i$, i.e. concentrate on the $i$th simplex, with each element of $\Delta_i^k$ denoted as a tuple $\mathbf{x}^i := (x_{i1}, x_{i2}, \ldots, x_{ik})$. From

---

[6]As before, in greater generality, it is possible to allow each simplex to be of different dimension and attain the same results. For simplicity of exposition, we restrict here to the special case in which all the simplices are of the same dimension.

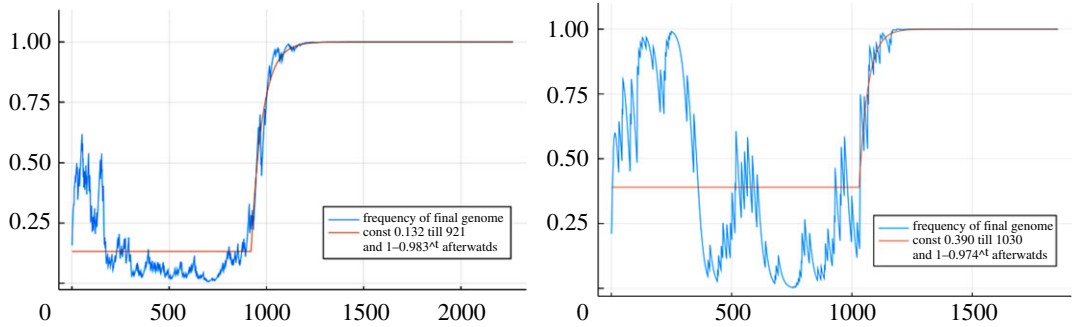

**Figure 1.** An illustration of the convergence to a monomorphic population consisting of one genotype under haploid sexual reproduction. In both simulations, the frequency of the final genome in the population over generations is tracked. This frequency appears to drift most of time, then converges at an exponential rate to fixation towards the end, in accordance with proposition 5.6.

the perspective of $\Delta_i^k$, $U$ may be considered to be a 'potential' function, involving $(x_{i1}, x_{i2}, \ldots, x_{ik})$ and other parameters.

Consider a Euclidean gradient vector derived from the potential in this perspective, that is, $\nabla U(\mathbf{x}^i) = (\partial U/\partial x_{i1}, \partial U/\partial x_{i2}, \ldots, \partial U/\partial x_{ik})$. Then the transformation of equation (A 1) can be considered as mapping $\mathbf{x}^i$ to $\nabla U(\mathbf{x}^i) \cdot \mathbf{x}^i$ for each $i$ separately, followed by projection to the simplex by way of the normalization. In a sense, the Baum–Eagon dynamic is an application of a form of 'gradient hill climbing', locally within each simplex of the polytope $\Theta$, that taken together ensures a global climb.

# Appendix B. Proofs

*Proof of proposition 4.1.* This is a straightforward application of Birkhoff's ergodic theorem. Again we register the log fitness. By the ergodic theorem, for each $g$, $\lim_{t\to\infty}(1/t)\ln(w_g^t(\omega)) = E(\ln w_g \mid \mu)$ with probability one. Hence the genotype $g^*$ with the greatest expected log fitness (which is also the one with the greatest expected fitness) dominates, as it grows at the fastest average rate. ∎

*Proof of theorem 4.4.* Consider the field $F_t^g = \sigma(\rho_1^g, \ldots, \rho_t^g)$ and the random variable

$$Z_t^g := \sum_{s=1}^t \rho_s^g - \sum_{s=1}^t E[\rho_{s+1}^g \mid F_t^g]. \tag{B1}$$

This is a martingale since

$$E[Z_{t+1}^g - Z_t^g \mid F_t^g] = E[\rho_{t+1}^g \mid F_t^g] - E[\rho_{t+1}^g \mid F_t^g] = 0, \tag{B2}$$

and it is bounded by assumption. Thus, by the Azuma–Hoeffding inequality:

$$\Pr(|t^{-1}Z_t^g| \geq ct^{-1/4}) \leq \exp\left(-\frac{c^2 t^{3/2}}{c^2 t}\right) = \exp(-\sqrt{t}). \tag{B3}$$

Let $E_t$ be the event that $|t^{-1}Z_t^g| \geq ct^{-1/4}$. Then we have shown that

$$\sum_{t=1}^\infty \Pr(E_t) \leq \sum_{t=1}^\infty \exp(-\sqrt{t}) < \infty. \tag{B4}$$

Using the Borel–Cantelli Lemma, we deduce

$$\Pr\left(\bigcap_{s\geq 1}\bigcup_{t\geq s} E_t\right) = 0. \tag{B5}$$

Note that

$$\bigcap_{s\geq 1}\bigcup_{t\geq s} E_t = \{|t^{-1}Z_t^g| \geq ct^{-1/4} \ \forall t \geq 1\}, \tag{B6}$$

implying that almost surely $(1/t)Z_t^g \to 0$ as $t \to \infty$. By the definition of $Z_t^g$, we deduce that almost surely

$$\lim_{t \to \infty} \left( \frac{1}{t} \sum_{s=1}^{t} \rho_s^g - \frac{1}{t} \sum_{s=1}^{t} E[\rho_{s+1}^g \mid F_s^g] \right) = 0. \tag{B7}$$

Thus, by adding $\limsup_{t \to \infty}(1/t) \sum_{s=1}^{t} E[\rho_{s+1}^g \mid F_s^g] = \limsup_{t \to \infty} \hat{\rho}_t^g$ to both sides of the equals sign in equation (B 7), we obtain that almost surely

$$\limsup_{t \to \infty} \frac{1}{t} \sum_{s=1}^{t} \rho_s^g \leq \limsup_{t \to \infty} \hat{\rho}_t^g, \tag{B8}$$

and by an entirely similar argument

$$\liminf_{t \to \infty} \frac{1}{t} \sum_{s=1}^{t} \rho_s^g \geq \liminf_{t \to \infty} \hat{\rho}_t^g. \tag{B9}$$

Next, recall that by assumption $g$ is asymptotically one-step-ahead superior on average, meaning that by definition $\liminf \hat{\rho}_t^g > \limsup \hat{\rho}_t^h$ for all genotypes $h \neq g$. Combining this with the inequalities in (B 8) and (B 9), which holds for every genotype, one obtains that for every genotype $h$

$$\liminf_{t \to \infty} \frac{1}{t} \sum_{s=1}^{t} \rho_s^g > \limsup_{t \to \infty} \frac{1}{t} \sum_{s=1}^{t} \rho_s^h. \tag{B10}$$

This is sufficient to deduce the statement of the theorem. ∎

*Proof of corollary 4.5.* As $\Pr(A_g^\infty) > 0$ by assumption, we can consider the process $\psi_t^g$ which is the restriction of $\psi_t$ to $A_g^\infty$. The corollary then follows by applying theorem 4.4 to the process $\psi_t^g$. ∎

*Proof of theorem 4.6.* As shown in [13], the replicator is an instantiation of Hedge, the exponential version of the multiplicative weights update algorithm. It follows that the replicator attain asymptotic zero regret.

Translating this mathematical result back to the evolutionary setting, this is equivalent to stating that asexual reproduction virtually converges monomorphically to an optimal-in-hindsight genotype $g^*$. ∎

*Proof of lemma 5.1.* Focus on a particular allele $a_{ij_i}$ and its attendant $q_{ij_i}$. Recall that by equation (2.2), $\bar{w} = \sum_{g \in \Gamma} d_g w_g$, that by equation (2.7), $w_{ij_i} = \sum_{g \in C_{a_{ij_i}}} w_g d_{g,i_{j_i}}$, while by equation (5.1) the haploid sexual replicator is $q_{ij_i}^+ = q_{ij_i}(w_{ij_i}/\bar{w})$.

For $g \notin C_{a_{ij_i}}$, one has $(\partial(d_g w_g))/\partial q_{ij_i} = 0$. For $g \in C_{a_{ij_i}}$, using $d_g = q_{1j_1}q_{2j_2}\ldots q_{mj_m}$ (by linkage equilibrium) yields

$$q_{ij_i} \frac{\partial(d_g w_g)}{\partial q_{ij_i}} = q_{1j_1}q_{2j_2}\ldots q_{mj_m} w_g = d_g w_g.$$

Hence $\partial \bar{w}/\partial q_{ij_i} = \sum_{g \in C_{a_{ij_i}}} w_g d_{g,ij_i} = w_{ij_i}$. It follows that $q_{ij_i}^+ = q_{ij_i}(\partial \bar{w}/\partial q_{ij_i})/\bar{w}$. This is the schema for applying the Baum–Eagon theorem of equation (A 1), with $T_1$ as the transformation and $\bar{w}$ the Lyapunov function. ∎

*Proof of theorem 5.2.* By lemma 5.1, monotonic mean fitness increase along trajectories is immediate from the Baum–Eagon Theorem.

Any pure strategy profile of the game $W_\Gamma$ (corresponding to a point distribution concentrated on a single allele at each locus) constitutes a fixed point of equation (2.8), and in fact the only fixed points of this dynamic are pure strategy profiles. However, if $g$ is a pure strategy profile of $w$ that is not a Nash equilibrium then it is not a stable point of the dynamic; using standard dynamics arguments involving nullclines and separatrices, there exists around $g$ a neighbourhood such that any $\bar{w}$-increasing trajectory with initial point in the interior of that neighbourhood eventually leaves that neighbourhood.

In other words, there is a basin of repulsion around every such point; hence the dynamic cannot converge to non-Nash equilibria points. Convergence will therefore always be to a pure strategy Nash equilibrium point, i.e. a local maximum of the potential $\bar{w}$. ∎

*Proof of theorem 5.4.* This is more or less a corollary of theorem 5.2. For each pure Nash equilibrium $v \in N_{W_\Theta}$, let $B_v$ be the set of elements in $\Delta(\Gamma)$ that asymptotically converge to $1_{g_v}$. By definition, this forms an asymptotically stable basin of attraction. ∎

*Proof of proposition 5.5.* This is an almost direct application of theorem 3 from [2]. We can identify $H_\nu$ as follows: for any $\eta > 0$, let $V_\eta$ be the connected component in $\Delta(\Gamma)$ of $\{x \in \Delta(\Gamma) \mid w(x) > w(1_{g_\nu}) - \eta\}$ that contains $1_{g_\nu}$.

Let $\eta_0 > 0$ be the smallest real number such that $\bar{V}_{\eta_0}$, the closure of $V_{\eta_0}$, contains another critical point of $w$ in addition to $1_{g_\nu}$. Set $H_\nu = \bigcup_{\eta < \eta_0} V_\eta$. Theorem 3 of [2] now applies to $H_\nu$ to attain the conclusion. ∎

*Proof of proposition 5.7.* Suppose that at time $t$ the population has mean fitness $\bar{w}$. A sexually reproducing sub-population of weight $r$ is selected, whose mean fitness is $\bar{w}_r$, and the complementary asexual sub-population of weight $1 - r$ therefore has mean fitness $\bar{w}_{1-r}$, such that $\bar{w} = r\bar{w}_r + (1-r)\bar{w}_{1-r}$. At time $t+1$, the offspring population of the sexual reproducers has mean fitness $\bar{w}_r^+$, and the offspring population of the asexual reproducers has mean fitness $\bar{w}_{1-r}^+$.

Since both the $T_0$ and the $T_1$ transformations increase mean fitness (except at fixed points), it follows that $\bar{w}_r^+ > \bar{w}_r$ and $\bar{w}_{1-r}^+ > \bar{w}_{1-r}$. But $\bar{w}^+ = r\bar{w}_r^+ + (1-r)\bar{w}_{1-r}^+$, hence $\bar{w}^+ > \bar{w}$.

This argument relies on the $T_1$ increasing mean fitness monotonically, which rests on theorem 5.2, which in turn ultimately relies on theorem 3.1. In game theoretic terms, theorem 3.1 presumes independent choices of strategies on the parts of the players (as opposed to correlated strategies), hence an initial point of linkage equilibrium needs to be assumed. Starting from any such interior point, the trajectory will follow increasing mean fitness until it arrives at a local maximum, which will be a pure Nash equilibrium point. ∎

*Proof of theorem 5.8.* Suppose that at time $t$ the population has mean fitness $\bar{w}$. For each partition $\lambda \in \Lambda$, a sub-population of weight $r_\lambda$ of reproduction type $\lambda$ is selected, whose mean fitness is $\bar{w}_\lambda$ such that $\bar{w} = \sum_{\lambda \in \Lambda} r_\lambda \bar{w}_\lambda$. At time $t+1$, the offspring population of the $\lambda$-type reproducers has mean fitness $\bar{w}_\lambda^+$, with population mean fitness $\bar{w}^+ = \sum_{\lambda \in \Lambda} r_\lambda \bar{w}_\lambda^+$.

Using similar argumentation as in the proof of proposition 5.7, since the $T_\lambda$ transformations increase mean fitness (except at fixed points) for all partitions $\lambda$, it follows that $\bar{w}_\lambda^+ > \bar{w}_\lambda$ for all partitions $\lambda$. Hence $\bar{w}^+ > \bar{w}$.

Starting from any linkage equilibrium point, the trajectory will follow increasing mean fitness until it arrives at a local maximum, which will be a pure Nash equilibrium point. ∎

*Proof of theorem 5.9.* In this proof, we ask a different question from the usual questions of regret minimization: instead of asking whether an algorithm attains the same asymptotic rate as the best expert, we suppose that the algorithm converges to the rate of the best expert and ask whether that rate is equal to some exogenous rate.

Let $e^1$, $e^2$, … be any environment realization. Let $i$ be a locus. By equation (2.8), the alleles in locus $i$ are each implementing a replicator equation (with respect to the identical interests game they are playing against the alleles in the other loci). This implies that the reproductive dynamic internal to the locus follows a multiplicative weights updating algorithm with respect to the total environment payoffs, taking into account both external and internal environments. Hence, regret minimization applies to this dynamic and in the limit the locus attains the average growth rate it would have attained had it implemented the optimal fixed strategy-in-hindsight within $\Delta(A_i)$ at all times. In other words, the individual locus attains virtual convergence.

From here, the proof proceeds inductively. Suppose that under the true dynamic each locus $i$ exhibits the mixed strategy sequence $\mu^i = (\mu_i^1, \mu_i^2, \ldots)$, where $\mu_i^t \in \Delta(A_i)$ for each $t$. Furthermore, denote by $L$ the lim sup average growth rate payoff that is attained under the profile $(\mu^1, \mu^2, \ldots, \mu^m)$ of these strategy sequences (which is equal for each locus).

Let $a_1^*$ represent the pure strategy of locus 1 whose lim inf attains asymptotically zero regret, as in equation (3.1), i.e, that asymptotically does as well as $L$. Locus 2 can then take the perspective of facing an environment consisting of the external environment along with internal environment $(a_1^*, \mu_i^3, \ldots)$, and virtually attain the same payoff with fixed optimal-in-hindsight $a_2^*$, i.e. $(a_1^*, a_2^*, \mu_i^3, \ldots)$ virtually attains $L$.

By induction, the sequence $(a_1^*, a_2^*, \ldots, a_{j-1}^*, \mu_i^j, \ldots)$ attains $L$. At locus $j$, implement the optimal-in-hindsight $a_j^*$ against $(a_1^*, a_2^*, \ldots, a_{j-1}^*, \mu_i^{j+1}, \ldots)$ to attain payoff $L$ under $(a_1^*, a_2^*, \ldots, a_{j-1}^*, a_j^*, \mu_i^{j+1}, \ldots)$.

Continuing by induction, in this way eventually one concludes that population consisting entirely of the genotype $g^* = a_1^* a_2^* \ldots a_m^*$ is the optimal-in-hindsight genotype. ∎

*Proof of proposition 5.10.* Following the same reasoning as in previous proofs, consider the perspective of locus $i$. The payoff received by locus $i$ is equal to what it would gain if all the other loci were to play the pure strategy profile $q_{-i} = (a_1^*, \ldots, a_{i-1}^*, a_{i+1}^*, \ldots, a_m^*)$.

By assumption the external environment process selecting the realizations $W_{e^1}, W_{e^2}, \ldots$ is stationary and ergodic. Since the $q_{-i}$ profile is virtually pure and fixed throughout time, the internal

environment $w_{e^t}(\cdot; q_{-1})$ reflects the external environment and is similarly stationary and ergodic. Hence the virtual convergence of locus $i$, which is equivalent to that of an asexually reproducing population under the conditions of a stationary and ergodic environment process, is always to the same payoff. This holds equally true for all loci. ∎

*Proof of theorem 6.1.* The proof is the same as the proof of theorem 5.4. ∎

*Proof of proposition 6.2.* The proof is the same as the proof of proposition 5.5. ∎

*Proof of theorem 6.3.* By equation (2.12), the alleles in the locus are implementing a replicator equation (with respect to the symmetric potential game they are playing among themselves). This implies that the reproductive dynamic in the locus follows a multiplicative weights updating algorithm with respect to the total environment payoffs, taking into account both external and internal environments. Hence, regret minimization applies to this dynamic and in the limit the locus attains the average growth rate it would have attained had it implemented the optimal fixed strategy-in-hindsight within $\Delta(A)$ at all times. □

*Proof of theorem 6.4.* Let $e^1, e^2, \ldots$ be any environment realization. Let $i$ be a locus. By equation (2.12), the alleles in locus $i$ are each implementing a replicator equation (with respect to the symmetric potential game they are playing among themselves). This implies that the reproductive dynamic internal to the locus follows a multiplicative weights updating algorithm with respect to the total environment payoffs, taking into account both external and internal environments. Hence, regret minimization applies to this dynamic and in the limit the locus attains the average growth rate it would have attained had it implemented the optimal fixed strategy-in-hindsight within $\Delta(A_i)$ at all times. In other words, the individual locus attains virtual (possibly polymorphic) convergence.

From here the proof proceeds inductively as in the proof of theorem 5.9. Suppose that under the true dynamic each locus $i$ exhibits the mixed strategy sequence $\mu^i = (\mu_i^1, \mu_i^2, \ldots)$, where $\mu_i^t \in \Delta(A_i)$ for each $t$. Furthermore, denote by $L$ the lim sup average growth rate payoff that is attained under the profile $(\mu^1, \mu^2, \ldots, \mu^m)$ of these strategy sequences (which is equal for each locus).

Letting $s_1^*$ represent the fixed (possibly mixed) strategy of locus 1 whose lim inf attains asymptotically zero regret, as in equation (3.1), i.e, that asymptotically does as well as $L$. Locus 2 can then take the perspective of facing an environment consisting of the external environment along with internal environment $(s_1^*, \mu_i^3, \ldots)$, and virtually attain the same payoff with fixed optimal-in-hindsight $s_2^*$, i.e. $(s_1^*, s_2^*, \mu_i^3, \ldots)$ virtually attains $L$.

Continuing argument by induction, in this way eventually one concludes that population consisting entirely of the genotype $g^* = s_1^* s_2^* \ldots s_m^*$ is the optimal-in-hindsight genotype. ∎

# Appendix C. Proof of convergence in potential games under the polynomial multiplicative weights update algorithm

## C.1. Preliminary setup

Let $(A, u, \phi)$ be a potential game, where $A = A_1 \times \cdots \times A_m$ is the set of action profiles, $u: A \to \mathbb{R}^m$ the payoff function, and $\Phi: A \to \mathbb{R}$ the potential. For $x \in A$, we use $x_i$ to denote the action in $x$ of the player $i$ and $x_{-i}$ to denote the actions in $x$ of the players apart from player $i$. For simplicity, we will assume that $|A_i| = k$ uniformly for all players; the extension to the more general case is straightforward. Enumerating the elements of $A_i$, the $j$th action of player $i$ is $a_{i_j}$.

A mixed strategy of player $i$ will be denoted $q_i = (q_{i_1}, \ldots, q_{i_k})$, and a profile of strategies $q = (q_1, \ldots, q_m) \in \Delta(A_1) \times \cdots \times \Delta(A_m)$. The application of a profile $q$ yields an expected payoff for player $i$ that we will denote $u_i(q)$. Given a pure action profile $x = (x_{1j_1}, x_{2j_2}, \ldots, x_{mj_m}) \in A$ and a profile of strategies $q$, denote

$$q_x = \prod_{1 \le \nu \le m} q_{\nu_{j_\nu}} \tag{C 1}$$

and

$$q_{x_{-i}} = \prod_{1 \le \nu \le m; \nu \ne i} q_{\nu_{j_\nu}}. \tag{C 2}$$

Suppose that each player is applying the multiplicative updates algorithm to update the mixed strategy he uses from one time period to the next. To interpret what is meant by this, we need to specify the payoff player $i$ receives for placing weight $q_{i_j}$ on action $a_{i_j}$ when his overall expected payoff is $u_i(q)$.

Suggestively borrowing notation introduced earlier here in the context of alleles, denote $C_{a_{ij}} = \{x \in A \mid x_i = a_{ij}\}$, i.e. the set of action profiles with the action of player $i$ fixed at $a_{i_j}$. Next suppose that player $i$ fixes action $a_{i_j}$ while the other players choose mixed strategies $q_{-i}$. In this case, denote the expected payoff for player $i$ by $u_{i_j}(q_{-i})$, which is

$$u_{i_j}(q_{-i}) := \sum_{x \in C_{a_{ij}}} q_{x_{-i}} u_i(x). \tag{C 3}$$

Denote by $u_{i_j}(q_{i_j}, q_{-i})$ the payoff player $i$ receives for placing weight $q_{i_j}$ on action $a_{i_j}$ when the other players choose $q_{-i}$. Using equation (C 3), this is

$$u_{i_j}(q_{i_j}, q_{-i}) := q_{i_j} u_{i_j}(q_{-i}). \tag{C 4}$$

With that we can specify what it means for each player to apply the multiplicative weights updates algorithm for $\eta > 0$. When $q$ is the profile of mixed strategies, player $i$ views the tuple $(u_{i_1}(q_{i_1}, q_{-i}), \ldots, u_{i_k}(q_{i_k}, q_{-i}))$. In response, the mixed strategy that player $i$ chooses in the next time period is given by

$$T(q_{i_j}) := q_{i_j} \frac{1 + \eta u_{i_j}(q_{-i})}{\sum_h q_{i_h}(1 + \eta u_{i_h}(q_{-i}))} \tag{C 5}$$

For ease of reading, we will from here express equation (C 5) more simply as

$$T(q_{i_j}) \propto q_{i_j}(1 + \eta u_{i_j}(q_{-i})), \tag{C 6}$$

supressing the denominator whose entire purpose is only to ensure that the result is a normalized probability distribution.

## C.2. Proof of theorem 3.1

We first prove that the potential payoff increases monotonically under the MWU algorithm in the special case that the potential game is an identical interests game and that the updating rule is the parameter-free case, i.e. for each $i, j$, $u_i(x) = u_j(x) = \Phi(x)$, so that each player gets the same payoff (the potential) for each profile of actions, and equation (C 6) becomes

$$T(q_{i_j}) \propto q_{i_j} u_{i_j}(q_{-i}). \tag{C 7}$$

From here most of the work is unravelling of definitions. From equation (C 3) and (C 4) and $u_i(x) = \Phi(x)$, we obtain

$$u_{i_j}(q_{-i}) = \sum_{x \in C_{a_{ij}}} q_{x_{-i}} \Phi(x). \tag{C 8}$$

At the same time, the expected payoff of player $i$ under $q$ is $\Phi(q) = \sum_{x \in A} q_x \Phi(x)$. It follows that for each available action $a_{i_j}$,

$$\frac{\partial \Phi}{\partial q_{i_j}} = \sum_{x \in C_{a_{ij}}} q_{x_{-i}} \Phi(x). \tag{C 9}$$

Putting it all together yields

$$T(q_{i_j}) \propto q_{i_j} \frac{\partial \Phi}{\partial q_{i_j}},$$

which is exactly what is needed for application of the Baum–Eagon Theorem (since $\Phi(q)$ is a polynomial function of the various probability weights of $q$). It follows that under this dynamic the value of $\Phi$ increases monotonically from one time period to the next as long as $T(q) \neq q$, hence there is convergence to a fixed point.

Moving on from the parameter-free case, consider next the more general polynomial update rule

$$T(q_{i_j}) \propto q_{i_j}(1 + \eta u_{i_j}(q_{-i})) = q_{i_j} + q_{i_j}(\eta u_{i_j}(q_{-i})),$$

but still maintain the assumption of an identical interest game, i.e. $u_i(x) = u_j(x) = \Phi(x)$ for all $i, j$, so that equation (C 8) still holds.

Define

$$\Psi(q) = \sum_{1 \leq i \leq m} \sum_{1 \leq j \leq k} q_{i_j} + \eta\Phi(q).$$

Then

$$\frac{\partial \Psi}{\partial q_{i_j}} = 1 + \eta\frac{\partial \Phi}{\partial q_{i_j}}, \tag{C 10}$$

while equation (C 9) holds as before, i.e. $\partial\Phi/\partial q_{i_j} = \sum_{x \in C_{a_{ij}}} q_{x_{-i}}\Phi(x)$. Putting it all together yields

$$T(q_{i_j}) \propto q_{i_j}\frac{\partial \Psi}{\partial q_{i_j}} = q_{i_j} + q_{i_j}(\eta u_{i_j}(q_{-i})),$$

which is sufficient for obtaining the result we seek by appeal to the Baum–Eagon Theorem.

Finally, in greatest generality suppose that the game is fully a potential game as opposed to an identical interests game. In this case, it is well known that the potential game can be decomposed into an identical interests game and a dummy game. That is, $u_i(q) = \Phi(q) + D_i(q_{-i})$, where $\Phi$ is an identical interests game and $D_i$ the payoff to $i$ from the dummy game $D$, depends solely on $q_{-i}$ but does not change at all with changes in $q_i$.

It follows that $\partial u_i/\partial q_{i_j} = \partial\Phi/\partial q_{i_j}$: in other words, this case essentially reduces to the case of an identical interests game from the perspective of partial differentiation with respect to the weights of the actions of player $i$. Hence, with minor modifications the same proof as applied earlier applies to the general case; we omit the obvious details.

Up to here, we have shown that in a potential game in which all players implement an MWU algorithm in discrete time all trajectories converge to a fixed point of the dynamic. Now we seek to show that the convergence is specifically to a Nash equilibrium. In §D.1, we show that in the special case of haploid reproduction under linkage equilibrium that around each point that is not a Nash equilibrium there is a basin of repulsion, hence convergence can only be to a Nash equilibrium.

We show here, using proof by contradiction, that this holds true in general potential games under the MWU dynamic. That is, if $q$ is an initial profile, we seek to ascertain whether the limit $p = \lim_{n \to \infty} T^n(q)$ can or cannot be different from a Nash equilibrium. Note that since $p$ is a fixed point of the dynamic, for each $i$, the support of $p_i$ is entirely on one action. Hence we can denote $p = (a_1^*, \ldots, a_m^*)$, where $a_i^* \in A_i$.

Suppose that $p$ is not a Nash equilibrium. Then there is an $i$ and an action $a'_i \in A_i$ such that $a'_i \neq a_i^*$ and $p' = (a_1^*, \ldots, a'_i, \ldots a_m^*)$ satisfies $\Phi(p') > \Phi(p)$. By continuity, there is an open set $O \subset \Delta(A_1) \times \cdots \times \Delta(A_m)$ containing $p$ such that for any $r = (r_1, \ldots, r_m) \in O$, one has for $r' = (r_1, \ldots, a'_i, \ldots, r_m)$ that $\Phi(r') > \Phi(r)$.

Now, since we assumed that $p = \lim_{n \to \infty} T^n(q)$, there is an integer $n_0$ such that for all $m \geq n_0$, $T^m(q) \in O$. Hence for each such $T^m(q)$, replacing the $i$th coordinate by the pure action $a'_i$ results in higher potential payoff. As detailed in the explanation in §3.2, this implies that under the MWU dynamic the weight of $a'_i$ can only increase at each time step from $T^m(q)$ to $T^{m+1}(q)$ for all $m \geq n_0$. This implies that the weight of $a'_i$ at $p = \lim_{n \to \infty} T^n(q)$ is positive, in contradiction to the assumption that $p$ plays pure strategy $a_i^*$.

# Appendix D. Basins of attraction and repulsion

It is clear from inspection of equation (2.8) that any pure strategy profile is a fixed point of the dynamic (and conversely the only fixed points are pure strategy profiles); the set of pure Nash equilibria is a proper subset of the set of pure strategy profiles. We complete this picture here by showing that around any pure strategy profile that is not a Nash equilibrium there is a basin of repulsion; conversely, around any profile that is a Nash equilibrium there is a basin of attraction.

We will make use of a collection of relative distribution weights (relative to an allele $a_{ij}$) $\{e^t_{g,ij}\}_{g \in C_{a_{ij}}}$. To define this, let $e^t_{g,ij} = 0$ if $g \notin C_{a_{ij_i}}$. Otherwise, define

$$e^t_{g,ij} := \frac{d^t_g}{\sum_{g' \in C_{a_{ij_i}}} d^t_{g'}}. \tag{D 1}$$

In words, $e^t_{g,ij}$ is the weight of gentotype $g$ among the genotypes containing $a_{ij}$. Clearly, $\sum_{g \in C_{a_{ij}}} e^t_{g,ij} = 1$.

When linkage equilibrium holds, we can re-express $e^t_{g,ij_i}$ associated with allele $a_{ij_i}$ when $g = a_{1j_1} a_{2j_2} \ldots a_{mj_m}$ as

$$e^t_{g,ij} = \frac{d^t_g}{\sum_{g' \in C_{a_{ij_i}}} d^t_{g'}} = \frac{q^t_{1j_1} q^t_{2j_2} \cdots q^t_{mj_m}}{\sum_{g' \in C_{a_{ij_i}}} d^t_{g'}}. \tag{D2}$$

Equation (2.7), defining the marginal fitness of allele $a_{ij}$ at time $t$ is rewritten in these terms as

$$w^t_{ij} := \sum_{g \in C_{a_{ij}}} w^t_g e^t_{g,ij}. \tag{D3}$$

It will be useful to express equation (D3) solely in terms of $\{q_i\}$ and $\{w_g\}$. To that end, for a fixed $a_{ij_i} \in \mathcal{A}_i$, enumerate the elements of $C_{a_{ij_i}}$ as $(g_1, \ldots, g_l)$. Each such $g \in C_{a_{ij_i}}$ is by definition a string of alleles $a_{1j_1} a_{2j_2} \ldots a_{ij_i} \ldots a_{mj_m}$ where $a_{ij_i}$ is the same for each $g \in C_{a_{ij_i}}$ but the other alleles vary from one such genotype to the other.

From equation (2.6), under linkage equilibrium, for each such $g$ we have $d^t_g = \prod_{\nu=1}^{m} q^t_{\nu j_\nu} = q^t_{1j_1} q^t_{2j_2} \cdots q^t_{ij_i} \cdots q^t_{mj_m}$. For future reference, we will want to use a 'reduced form' of this expression, defined as

$$d^t_{g,-i} = \prod_{1 \le \nu \le m, \nu \ne i} q^t_{\nu j_\nu} = q^t_{1j_1} q^t_{2j_2} \cdots q^t_{ij_i} \cdots q^t_{mj_m} \tag{D4}$$

by which we mean the $m-1$-fold product that does not include $q^t_{ij}$ in it. Then we can re-write equation (D2) as

$$\begin{aligned} e^t_{g,ij_i} &= \frac{d^t_g}{\sum_{g' \in C_{a_{ij_i}}} d^t_{g'}} = \frac{q^t_{1j_1} q^t_{2j_2} \cdots q^t_{mj_m}}{\sum_{g' \in C_{a_{ij_i}}} d^t_{g'}} \\ &= q^t_{ij_i} \frac{d^t_{g,-i}}{\sum_{g' \in C_{a_{ij_i}}} d^t_{g'}} = \frac{q^t_{ij_i}}{q^t_{ij_i}} \frac{d^t_{g,-i}}{\sum_{g' \in C_{a_{ij_i}}} d^t_{g',-i}} \\ &= \frac{d^t_{g,-i}}{\sum_{g' \in C_{a_{ij_i}}} d^t_{g',-i}}. \end{aligned} \tag{D5}$$

Note that this entirely removes dependence on $q^t_{ij_i}$; all dependence is on the allelic frequencies of loci *apart* from locus $i$. We can go even further. Denote in vector notation $\vec{d}^t_{ij_i,-i} := (d^t_{g_1,-i}, \ldots, d^t_{g_l,-i})$, with respect to the eumeration of $(g_1, \ldots, g_l)$ as the elements of $C_{a_{ij_i}}$. By tracing through the definitions, it becomes clear that the dependence on $a_{ij_i}$ is superfluous. In other words, for $a_{ij_i}, a_{ij'_i} \in \mathcal{A}_i$, two alleles in locus $i$, one has $\vec{d}^t_{ij_i,-i} = \vec{d}^t_{ij'_i,-i}$. We can, therefore, denote this vector uniformly as $\vec{d}^t_{-i}$.

Similarly, $\sum_{g \in C_{a_{ij_i}}} d^t_{g,-i} = \sum_{g \in C_{a_{ij'_i}}} d^t_{g,-i}$ for $a_{ij_i}, a_{ij'_i} \in \mathcal{A}_i$. Hence we can choose any one of them and uniformly denote $N_{-i} := \sum_{g \in C_{a_{ij_i}}} d^t_{g,-i}$.

Continuing with the vector notation, denote $\vec{w}_{ij_i} := (w_{g_1}, \ldots, w_{g_l})$ with respect to the enumeration of $(g_1, \ldots, g_l)$ as the elements of $C_{a_{ij_i}}$. In this, we cannot avoid dependence on the specific allele $a_{ij_i}$.

From here, we can rewrite equation (D3) as

$$w^t_{ij_i} = \sum_{g \in C_{a_{ij_i}}} w^t_g e^t_{g,ij_i} = \frac{1}{N_{-i}} \langle \vec{d}^t_{-i}, \vec{w}_{ij_i} \rangle \tag{D6}$$

using the vector dot product.

Finally, since the mean fitness $\bar{w}^t_i = \sum_{j=1}^{k} q^t_{ij} w^t_{ij}$, this becomes

$$\bar{w}^t_i = \frac{1}{N_{-i}} \sum_{j=1}^{k} q^t_{ij} \langle \vec{d}^t_{-i}, \vec{w}_{ij} \rangle. \tag{D7}$$

## D.1. Basins of repulsion around non-equilibrium points

Suppose that fitness values $w_g$ are fixed over time. Let $(a^*_1, a^*_2, \ldots, a^*_m)$ be a pure strategy profile of the game that is not a Nash equilibrium. In the biological interpretation, this corresponds to a population composed solely of genotypes $g = a^*_1, a^*_2, \ldots, a^*_m$, in which case $\bar{w} = w_g = w(a^*_1, a^*_2, \ldots, a^*_m)$.

By definition of a non-equilibrium, there is at least one player/locus $\bar{i}$ such that, if we write $a^*_i = a_{\bar{i}j}$ then there is $a_{ij'}$ such that $w(a_{ij'}; a^*_{-\bar{i}}) > w(a^*_i; a^*_{-\bar{i}})$. For $\varepsilon > 0$, construct for each player $i$ the mixed strategy $q^\varepsilon_i$ that gives weight $1 - \varepsilon$ to pure action $a^*_i \in \mathcal{A}_i$ and spreads the rest of the weight $\varepsilon$ among all the other actions in $\mathcal{A}_i$.

We claim that for sufficiently small $\varepsilon$, the haploid sexual replicator dynamic at the mixed strategy $(q_1^\varepsilon, \ldots, q_m^\varepsilon)$ draws away from $(a_1^*, a_2^*, \ldots, a_m^*)$.

To see this, note that for $(q_1^\varepsilon, \ldots, q_m^\varepsilon)$ for small $\varepsilon$, for any $i$ and $ij_i \in \mathcal{A}_i$, it is the case that $w_{ij_i}$ is very close to $w(a_{ij_i}; a_{-i}^*)$. This is because by equation (D 6),

$$w_{ij_i} = \frac{1}{N_{-i}} \langle \vec{d}_{-i}^t, \vec{w}_{ij_i} \rangle,$$

where each element $d_{g,-i} \in \vec{d}_{-i}$, for $g = a_{1j_1} \ldots a_{ij_i} \ldots a_{mj_m}$, is defined by $d_{g,-i} = \prod_{1 \le v \le m, v \neq i} q_{vj_v}^\varepsilon$. By the definition of $(q_1^\varepsilon, \ldots, q_m^\varepsilon)$, for small $\varepsilon$ the only relevant $d_{g,-i}$ that has appreciable weight is that associated with $g = a_1^* \ldots a_{ij_i} \ldots a_m^*$, where $d_{g,-i} = \prod_{1 \le v \le m, v \neq i} (1 - \varepsilon)$. Hence $w_{ij_i}$ is very nearly $w(a_{ij_i}; a_{-i}^*)$.

In particular, $w_{ij'} \approx w(a_{ij'}; a_{-\bar{i}}^*) > w(a_{\bar{i}}^*; a_{-\bar{i}}^*) \approx \varphi_{i^*}$. By continuity, for sufficiently small $\varepsilon$ this strict inequality can be guaranteed to hold, $w_{ij'} > w_{i^*}$.

Since the players are implementing the haploid sexual replicator equation, equation (2.8), in updating from one period to the next, when $w_{ij'} > w_{i^*}$ at time $t$ it follows that at time $t + 1$ in the distribution $q_i^{t+1}$ relatively greater weight will be given to $a_{ij'}$ and relatively less to $a_i^*$. It follows that in the step from $t$ to $t + 1$ the dynamic moves away from $(a_1^*, a_2^*, \ldots, a_m^*)$.

This is sufficient to conclude that the dynamic must converge to a pure Nash equilibrium.

## D.2. Proof of proposition 5.6

Suppose that $a^* = (a_{1j_1^*} a_{2j_2^*} \ldots a_{mj_m^*})$ is a pure strategy Nash equilibrium profile of the game.

Let $B \subset D$ be defined as the collection of mixed strategy profiles $(q_1, \ldots, q_m) \in D$ such that for all $i$:

(1) $q_{ij_i^*} > q_{ij_i}$ for all $j_i \neq j_i^* \in \mathcal{A}_i$, and
(2) $w_{ij_i^*} > w_{ij_i}$ for all $j_i \neq j_i^* \in \mathcal{A}_i$.

In words, $B$ is the set of mixed strategy profiles such that for each $i$, within the allelic frequency distribution $q_i$ the greatest weight is placed on $a_{1j_1^*}$ and at the same time the marginal fitness of $a_{1j_1^*}$ is the highest against all its competing alleles in locus $i$. (Recall that by equation (2.7) $w_{ij_i}$ is a function of genotypic density and hence also a function of $(q_1, \ldots, q_m)$.)

We can now proceed to show that $B$ is an exponentially stable basin of attraction around $a^*$. To see this, first note that $B$ is not empty because the profile $a^*$ is trivially an element of $B$. By continuity, $B$ is then a neighbourhood of $a^*$.

Next, let $(q_1^t, \ldots, q_m^t) \in B$ at time $t$. For any $i$, since $w_{ij_i^*} > w_{ij_i}$ for all $j_i \neq j_i^*$, by equation (2.8) it follows that at $t + 1$ the weight of allele $a_{ij_i^*}$, i.e. $q_{ij_i^*}^{t+1}$, relative to $q_{ij_i}^{t+1}$ for any other allele, only increases. So the first condition for $q_{ij_i}^{t+1}$ is satisfied.

Next, since $a^*$ is a Nash equilibrium, $w(a^*) > w(a_{ij_i}; a_{-i^*})$ for all $j_i \neq j_i^*$. Recall that $w_{ij_i}^{t+1} = (1/N_{-i}) \langle \vec{d}_{-i}^{t+1}, \vec{w}_{ij_i} \rangle$. Since the weight $q_{i'j_{i'}^*}^{t+1}$ increases relative to $q_{i'j_{i'}}^{t+1}$ for all $i'$, and $w(a_{ij_i^*}; a_{-i^*}) > w(a_{ij_i}; a_{-i^*})$ for all $j_i \neq j_i^*$, it can only be the case that $\langle \vec{d}_{-i}^{t+1}, \vec{v}_{ij_i^*} \rangle > \langle \vec{d}_{-i}^{t+1}, \vec{v}_{ij_i} \rangle$, i.e. $w_{ij_i^*}^{t+1} > w_{ij_i}^{t+1}$. Hence the second condition is satisfied and it follows that $(q_1^{t+1}, \ldots, q_m^{t+1}) \in B$.

Finally, since for each $i$ the weight $q_{ij_i^*}$ increases relative to any other $q_{ij_i}$ monotonically (and even at an increasing rate), asymptotically starting from any $(q_1, \ldots, q_m) \in B$ the dynamic converges to $a^*$.

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
