## [Peer Review File · Royal Society Open Science]

Review History

RSOS-210309.R0 (Original submission)

Review form: Reviewer 1

Is the manuscript scientifically sound in its present form?

Yes

Are the interpretations and conclusions justified by the results?

Yes

Is the language acceptable?

Yes

Do you have any ethical concerns with this paper?

No

Have you any concerns about statistical analyses in this paper?

No

Recommendation?

Major revision is needed (please make suggestions in comments)

Comments to the Author(s)

Making the Most of Potential: Potential Games and Genotypic Convergence

I think this is a very interesting work and can be considered for publication after the following concerns are addressed:

1. The references can be given in Figure 1 corresponding to each result. In addition, should it be a Table rather than a Figure?
2. What is the meaning of d_g^+ in Equation 4? What is its relationship to Equation 3?
3. There are several expressions of d_i^+ in this manuscript, such as Eq. (17), the second equation on Page 17, and the second equation on Page 18. What are the difference and their application in the model of this manuscript?
4. Why do the authors set w_g^t and w_h^t as this form ($e^{1/3}$ and $e^{1/2}$) on page 20? The author can explain this setting or include some references.
5. Are there more examples to show the effectiveness of this proposed model? The authors only gave 2 examples in Table 1.
6. The discussion should be elaborated and Conclusion Section should be given too.
7. on a side note, there seems to be some overlap with the results from:
<https://doi.org/10.1002/advs.202001995>
<https://doi.org/10.1038/srep34889>
 If so, do discuss these two papers in the introduction /background to motivate your work further.

Review form: Reviewer 2

Is the manuscript scientifically sound in its present form?

Yes

Are the interpretations and conclusions justified by the results?

Yes

Is the language acceptable?

Yes

Do you have any ethical concerns with this paper?

No

Have you any concerns about statistical analyses in this paper?

No

Recommendation?

Accept with minor revision (please list in comments)

Comments to the Author(s)

Summary:

In this paper, the authors study evolutionary convergence for different kinds of reproduction (asexual, haploid, diploid). They do so by interpreting the dynamics as the evolutionary

dynamics of a (potential) game, and by applying insights from game theory (regret minimization) and by using the Baum Eagon theorem. The main results are as follows:

(-) If the fitness of genotypes is fixed (i.e., there is a constant environment), and if reproduction is asexual, haploid, or single-locus diploid, the authors prove that the dynamics will always converge to a fixed point (which is shown to be a Nash equilibrium).

(-) In all other cases, the dynamics converges "virtually". This means that the dynamics itself may not converge (which is intuitively clear); however, the reproducing population asymptotically attains the same mean growth rate that it would have obtained, had it been comprised by the ex-post optimal genotype.

General evaluation:

The article is absolutely impressive. It combines techniques from different fields (population genetics, algorithms, game theory) to propose an elegant framework for how to analyse convergence of evolving populations. The introduction is extremely well written and provides a good summary of what the authors aim to do (especially the table in Figure 1 is very helpful). In the following, I only have a few smaller comments that the authors may take into account to further improve the paper.

Major Comment:

(-) Maybe the biggest advantage of the paper, its multidisciplinary, also entails a certain risk. There are probably not too many scholars that have sufficient knowledge in both population genetics and (algorithmic) game theory to appreciate all of the papers' findings. I do believe that the authors do a very good job in explaining the concepts they apply. However, in a few cases, some further exposition would have been useful.

For example, I'd appreciate if the authors could explain in more detail, perhaps in Section 1.2, what virtual convergence is good for. What are the precise conceptual insights it offers? Can an it be used as a tool to prove further statements of interest?

Similarly, what is the intuition behind a potential function? How does it help to prove things? Moreover, I'd appreciate an explicit statement about which results are already known (although the existing proof may be based on different techniques). For example, I'd assume that all non-virtual convergence results are already known, is this correct?

Given the many variables (and the relations between them), it might be useful to have a table that contrasts the variables (and interpretations) in game theory and the variables (and interpretations) in population genetics.

Finally, I'd appreciate if there was a short conclusions section that (once more) summarises the results and puts them into the perspective of the previous literature.

Minor Comments:

(-) Last paragraph Section 1.1: "relatively small increased" -> "relatively small increases"

(-) First paragraph Section 2.5: The index for $\{w_g^t\}$ should be " $g \in \Gamma$ " instead of " $\omega \in \Gamma$ "

(-) Section 3: Could you please add some references to the relevant literature on regret minimisation, and to the multiplicative weights algorithm?

(-) Section 3.2: " l_i is greater (respectively less than) than..." -- omit the first "than"

(-) Proposition 8: Does this Proposition need an additional assumption? For example, in its present form, the Proposition seems wrong for pure Nash equilibria that are weakly dominated -- for example for the 2x2 symmetric payoff matrix where the first player's payoff is 1 1 (first row) and 1 2 (second row). Here, the first strategy is a Nash equilibrium, but it's not locally stable under replicator dynamics.

In the proof, the authors seem to assume the Nash equilibrium is strict.
(-) Section 6.1.1: "are generically monotonically increase" -> "increasing"

Decision letter (RSOS-210309.R0)

Dear Dr Edhan

The Editors assigned to your paper RSOS-210309 "Making the Most of Potential: Potential Games and Genotypic Convergence" have now received comments from reviewers and would like you to revise the paper in accordance with the reviewer comments and any comments from the Editors. Please note this decision does not guarantee eventual acceptance.

Please submit your revised manuscript and required files (see below) no later than 21 days from today's (ie 04-May-2021) date. Note: the ScholarOne system will 'lock' if submission of the revision is attempted 21 or more days after the deadline. If you do not think you will be able to meet this deadline please contact the editorial office immediately.

on behalf of Dr Derek Abbott (Associate Editor) and Mark Chaplain (Subject Editor)
openscience@royalsociety.org

Associate Editor Comments to Author (Dr Derek Abbott):

Comments to the Author:

Please provide a point by point response to all reviewer comments.

Reviewer comments to Author:

Reviewer: 1

Comments to the Author(s)

Making the Most of Potential: Potential Games and Genotypic Convergence

I think this is a very interesting work and can be considered for publication after the following concerns are addressed:

1. The references can be given in Figure 1 corresponding to each result. In addition, should it be a Table rather than a Figure?
2. What is the meaning of d_g^+ in Equation 4? What is its relationship to Equation 3?
3. There are several expressions of d_i^+ in this manuscript, such as Eq. (17), the second equation on Page 17, and the second equation on Page 18. What are the difference and their application in the model of this manuscript?
4. Why do the authors set w_g^t and w_h^t as this form ($e^{1/3}$ and $e^{1/2}$) on page 20? The author can explain this setting or include some references.
5. Are there more examples to show the effectiveness of this proposed model? The authors only gave 2 examples in Table 1.
6. The discussion should be elaborated and Conclusion Section should be given too.
7. on a side note, there seems to be some overlap with the results from:
<https://doi.org/10.1002/advs.202001995>
<https://doi.org/10.1038/srep34889>
 If so, do discuss these two papers in the introduction /background to motivate your work further.

Reviewer: 2

Comments to the Author(s)

Summary:

In this paper, the authors study evolutionary convergence for different kinds of reproduction (asexual, haploid, diploid). They do so by interpreting the dynamics as the evolutionary dynamics of a (potential) game, and by applying insights from game theory (regret minimization) and by using the Baum Eagon theorem. The main results are as follows:

- (-) If the fitness of genotypes is fixed (i.e., there is a constant environment), and if reproduction is asexual, haploid, or single-locus diploid, the authors prove that the dynamics will always converge to a fixed point (which is shown to be a Nash equilibrium).
- (-) In all other cases, the dynamics converges "virtually". This means that the dynamics itself may not converge (which is intuitively clear); however, the reproducing population asymptotically attains the same mean growth rate that it would have obtained, had it been comprised by the ex-post optimal genotype.

General evaluation:

The article is absolutely impressive. It combines techniques from different fields (population genetics, algorithms, game theory) to propose an elegant framework for how to analyse convergence of evolving populations. The introduction is extremely well written and provides a

good summary of what the authors aim to do (especially the table in Figure 1 is very helpful). In the following, I only have a few smaller comments that the authors may take into account to further improve the paper.

Major Comment:

(-) Maybe the biggest advantage of the paper, its multidisciplinaryity, also entails a certain risk. There are probably not too many scholars that have sufficient knowledge in both population genetics and (algorithmic) game theory to appreciate all of the papers' findings. I do believe that the authors do a very good job in explaining the concepts they apply. However, in a few cases, some further exposition would have been useful.

For example, I'd appreciate if the authors could explain in more detail, perhaps in Section 1.2, what virtual convergence is good for. What are the precise conceptual insights it offers? Can an it be used as a tool to prove further statements of interest?

Similarly, what is the intuition behind a potential function? How does it help to prove things? Moreover, I'd appreciate an explicit statement about which results are already known (although the existing proof may be based on different techniques). For example, I'd assume that all non-virtual convergence results are already known, is this correct?

Given the many variables (and the relations between them), it might be useful to have a table that contrasts the variables (and interpretations) in game theory and the variables (and interpretations) in population genetics.

Finally, I'd appreciate if there was a short conclusions section that (once more) summarises the results and puts them into the perspective of the previous literature.

Minor Comments:

(-) Last paragraph Section 1.1: "relatively small increased" -> "relatively small increases"

(-) First paragraph Section 2.5: The index for $\{w_g^t\}$ should be " $g \in \Gamma$ " instead of " $\omega \in \Gamma$ "

(-) Section 3: Could you please add some references to the relevant literature on regret minimisation, and to the multiplicative weights algorithm?

(-) Section 3.2: " l_i is greater (respectively less than) than..." -- omit the first "than"

(-) Proposition 8: Does this Proposition need an additional assumption? For example, in its present form, the Proposition seems wrong for pure Nash equilibria that are weakly dominated -- for example for the 2x2 symmetric payoff matrix where the first player's payoff is 1 1 (first row) and 1 2 (second row). Here, the first strategy is a Nash equilibrium, but it's not locally stable under replicator dynamics.

In the proof, the authors seem to assume the Nash equilibrium is strict.

(-) Section 6.1.1: "are generically monotonically increase" -> "increasing"

===PREPARING YOUR MANUSCRIPT===

===PREPARING YOUR REVISION IN SCHOLARONE===

<https://royalsociety.org/journals/authors/author-guidelines/#supplementary-material> to include a suitable title and informative caption. An example of appropriate titling and captioning may be found at https://figshare.com/articles/Table_S2_from_Is_there_a_trade-off_between_peak_performance_and_performance_breadth_across_temperatures_for_aerobic_sc_ope_in_teleost_fishes_/3843624.

Author's Response to Decision Letter for (RSOS-210309.R0)

See Appendix A.

RSOS-210309.R1 (Revision)

Review form: Reviewer 1

Is the manuscript scientifically sound in its present form?

Yes

Are the interpretations and conclusions justified by the results?

Yes

Is the language acceptable?

Yes

Do you have any ethical concerns with this paper?

No

Have you any concerns about statistical analyses in this paper?

No

Recommendation?

Accept with minor revision (please list in comments)

Comments to the Author(s)

Please proofread the manuscript carefully and check that the references are cited properly, before uploading. It is extremely difficult to read the revised manuscript as there is literally no point to point responses except for some very brief comments in the cover letter. The authors have also uploaded 4 sets of revised manuscript totaling 200 over pages, rendering the reading very difficult.

Review form: Reviewer 2

Is the manuscript scientifically sound in its present form?

Yes

Are the interpretations and conclusions justified by the results?

Yes

Is the language acceptable?

Yes

Do you have any ethical concerns with this paper?

No

Have you any concerns about statistical analyses in this paper?

No

Recommendation?

Accept as is

Comments to the Author(s)

The authors have taken all my major suggestions into account. In particular,

- (i) They explain now in more detail how the framework of potential games (and potential functions) is useful.
- (ii) They explain in more detail how their results add to the existing literature, and
- (iii) They now provide a summarizing Conclusions section.

I appreciate all these changes. In my opinion, already the first version of the manuscript was very good, and hence I support publication.

Just two minor comments:

- (-) I find this sentence a bit weird: "When the environment is fixed, they [reproductive processes] will converge to the stability of local optima as represented by Nash equilibria". I'd rather write "... they will converge to local optima represented by Nash equilibria"
- (-) On the same page, there is a "tuhs" that should be replaced by "thus"

Decision letter (RSOS-210309.R1)

Dear Dr Edhan

On behalf of the Editors, we are pleased to inform you that your Manuscript RSOS-210309.R1 "Making the Most of Potential: Potential Games and Genotypic Convergence" has been accepted for publication in Royal Society Open Science subject to minor revision in accordance with the referees' reports. Please find the referees' comments along with any feedback from the Editors below my signature.

Please submit your revised manuscript and required files (see below) no later than 7 days from today's (ie 23-Jul-2021) date. Note: the ScholarOne system will 'lock' if submission of the revision is attempted 7 or more days after the deadline. If you do not think you will be able to meet this deadline please contact the editorial office immediately.

on behalf of Dr Derek Abbott (Associate Editor) and Mark Chaplain (Subject Editor)
openscience@royalsociety.org

Reviewer comments to Author:

Reviewer: 1

Comments to the Author(s)

Please proofread the manuscript carefully and check that the references are cited properly, before uploading. It is extremely difficult to read the revised manuscript as there is literally no point to point responses except for some very brief comments in the cover letter. The authors have also uploaded 4 sets of revised manuscript totaling 200 over pages, rendering the reading very difficult.

Reviewer: 2

Comments to the Author(s)

The authors have taken all my major suggestions into account. In particular,

(i) They explain now in more detail how the framework of potential games (and potential functions) is useful.

(ii) They explain in more detail how their results add to the existing literature, and

(iii) They now provide a summarizing Conclusions section.

I appreciate all these changes. In my opinion, already the first version of the manuscript was very good, and hence I support publication.

Just two minor comments:

- (-) I find this sentence a bit weird: "When the environment is fixed, they [reproductive processes] will converge to the stability of local optima as represented by Nash equilibria". I'd rather write "... they will converge to local optima represented by Nash equilibria"
- (-) On the same page, there is a "tuhs" that should be replaced by "thus"

===PREPARING YOUR MANUSCRIPT===

===PREPARING YOUR REVISION IN SCHOLARONE===

Author's Response to Decision Letter for (RSOS-210309.R1)

See Appendix B.

Decision letter (RSOS-210309.R2)

Dear Dr Edhan,

I am pleased to inform you that your manuscript entitled "Making the Most of Potential: Potential Games and Genotypic Convergence" is now accepted for publication in Royal Society Open Science.

on behalf of Dr Derek Abbott (Associate Editor) and Mark Chaplain (Subject Editor)
openscience@royalsociety.org

Appendix A

May 2021

To: The Editor of Royal Society Open Science,

We thank you for the opportunity to revise again our manuscript, Making the Most of Potential, Potential Games and Genotypic Convergence, for re-submission to the journal Royal Society Open Science.

Here is a summary of the changes that we implemented:

- Figure 1 has been renamed Table 1.
- Table 1 now includes references to the theorems in the manuscript.
- In several places we now explain explicitly our use of notation such as d_g^+ or d_i^+ , in response to a question by reviewer 1.
- Reviewer 1 asked why we set w_g^t and w_h^t to be $e^{1/3}$ and $e^{1/2}$ on page 21. The answer is that this is just one example of a process of temporally varying environments that does not converge to a single genotype. It so happens that the example we chose involves terms of the form $e^{1/3}$ and $e^{1/2}$, to illustrate what may happen. There is nothing beyond that implied in the example.
- Reviewer 1 also asks about the examples in what is now Table 3 of the manuscript. We wish to emphasise that what appears in Table 3 is only an illustrative numerical ‘toy’ example, to aid us in making a point. It represents a possible hypothetical population that occasionally experiences ‘rainy years’ and at times ‘drought years’. There is no intention here to state a theorem or imply any direct applicability of the models of the manuscript.
- We have added more discussion in the introduction and a conclusions section, in response to a comment of Reviewer 1.
- Reviewer 1 mentioned some overlaps between the subjects covered in our work and those of Cheong et al., 2016, and Babajanyan et al., 2020. We have added a literature review section that now includes explicit reference to those research articles.
- Reviewer 2 asked if it might be possible to add more detail in Section 1.2 regarding the concept we introduce of virtual convergence, including conceptual insights. We have done so.
- Similarly, Reviewer 2 asked about the intuition behind a potential function. More on this concept has therefore been added in Section 1.1.
- We have added a literature review section in Section 1.4. This section includes explicit delineation of the results in our manuscript

that have already appeared in the literature (perhaps based on different techniques) versus those that are novel here.

- We have added a table (Table 2) that expressly presents and contrasts, side-by-side, the variables used in the game theoretic sections and the population genetics sections of the manuscript, for the benefit of readers.
- We have added a conclusions section (Section 7) that summarises the results and puts them into the perspective of the previous literature.
- We have corrected all of the typos spotted by the reviewers and others that were spotted by us in the course of the revision.

Thank you again for your consideration,
Sincerely,

Omer Edhan

Ziv Hellman

Ilan Nehama

Appendix B

July 2021

To: The Editor of Royal Society Open Science,

We thank you for accepting our manuscript subject to minor revisions. We thank the referees for their kind service and very helpful comments.

Here is a summary of the changes that we implemented:

- We have clarified the sentence in paragraph 3 of page 5. It now reads “When the environment is fixed, they will converge to local optima as represented by Nash equilibria.”
- We have corrected a typo in paragraph 4 of page 5. It now reads ‘thus’ and not ‘tuhs’.

Thank you again for your consideration,
Sincerely,

Omer Edhan

Ziv Hellman

Ilan Nehama